# A Closer Look at Changes in High-Risk Food-Handling Behaviors and Perceptions of Primary Food Handlers at Home in South Korea across Time

**DOI:** 10.3390/foods9101457

**Published:** 2020-10-13

**Authors:** Tae Jin Cho, Sun Ae Kim, Hye Won Kim, Min Suk Rhee

**Affiliations:** 1Department of Food and Biotechnology, College of Science and Technology, Korea University, 2511, Sejong-ro, Sejong 30019, Korea; microcho@korea.ac.kr; 2Department of Food Science and Engineering, Ewha Womans University, Seoul 03760, Korea; sunaekim@ewha.ac.kr; 3Department of Biotechnology, College of Life Sciences and Biotechnology, Korea University, 145, Anam-ro, Seongbuk-gu, Seoul 02841, Korea; kgpdnjs@korea.ac.kr

**Keywords:** consumer survey, food safety, food hygiene, food handling, consumer behavior, risk perception, healthy food consumption, cultural consumer context, microbiological risk, health

## Abstract

Food-handling behaviors and risk perceptions among primary food handlers were investigated by consumer surveys from different subjects in 2010 (N = 609; 1st survey will be called here “Year 2010”) and 2019 (N = 605; 2nd survey will be called here “Year 2019”). Year 2010 was characterized by consumers’ risk perception-behavior gap (i.e., consumers knew safe methods for food-handling, but responses regarding the behaviors did not support their confidence in food safety): they (1) did not wash/trim foods before storage, (2) thawed frozen foods at room temperature, and (3) exposed leftovers to danger zone temperatures. These trends were not improved and the gaps in Year 2010 remained in Year 2019. Year 2010 was also characterized by other common high-risk behaviors improved during 8 years for the following aspects: (1) 70.0% of consumers divided a large portion of food into smaller pieces for storage, but few consumers (12.5%) labeled divided foods with relevant information, and (2) they excessively reused kitchen utensils. Whereas in Year 2019, more consumers (25.7%) labeled food and usage periods for kitchen utensils were shortened. Consumers usually conformed to food safety rules in both Year 2010 and 2019: (1) separate storage of foods, (2) storage of foods in the proper places/periods, (3) washing fruits/vegetables before eating, (4) washing hands after handling potentially hazardous foods, and (5) cooking foods and reheating leftovers to eat. Our findings provided resources for understanding consumers’ high-risk behaviors/perceptions at home, highlighting the importance of behavioral control.

## 1. Introduction

Food safety is one of the most important global public health issues which has repeatedly created social anxiety and resulted in the economic loss of many countries [1,2,3]. Most people commonly know that foodborne diseases are mainly associated with foods consumed outside the home, however, the private home is also a crucial site where foodborne illnesses are engendered [4,5,6]. The World Health Organization (WHO) [7] estimated that approximately 40% of foodborne illnesses have been associated with food prepared at home, and Redmond and Griffith [8] also reported that 50–87% of foodborne diseases were attributed to the consumption of food at home. Researchers assumed that the number of actual foodborne diseases caused at home might be much higher than reported since most foodborne illnesses were unreported and/or unconfirmed [4]; Redmond and Griffith [9] estimated that this number could reach 95%. In South Korea, it is also assumed that the number of foodborne diseases has been underestimated (i.e., most cases are expected to be unreported); Ministry of Food and Drug Safety reported that only 1.32% outbreaks and 0.36% cases were attributed to foods prepared at private home [10].

Mishandling of foods could frequently occur from preparation, handling, cooking, and storing in the home [11]. Byrd-Bredbenner et al. [4] summarized the reasons why the home was a risky place for foodborne disease as follows: (1) the greatest portion of foods is prepared at home; (2) there are many consumers at home in high risk groups for health problems (YOPI: young/old/pregnant/immunocompromised people); (3) people even in the YOPI group do not perceive themselves as susceptible to illness or do not follow recommended practices for food safety, and (4) home-prepared food can be served to a wider community (e.g., school picnics, lunch boxes, and bake sales).

Risk perception on food hygiene has been regarded as one of the most important topics for estimating the actual risk levels of food safety for lay public [12]. Consumer surveys can be used to explore the underreported and/or underrecognized risk perceptions linked to the improper behaviors with the perspective to domestic food safety, which could also support to overcome the limitation of current conventional consumer guidelines [13,14]. Moreover, understanding national differences in risk perceptions should be the pre-requisites for the establishment of internationally-recognized interventions (e.g., the education, public campaign, etc.) against the major risk factors [15]. Especially previous research has focused on the gap between risk perception and actual practices of food handlers as the key clues for the improvement of both the proper perception and behaviors for food safety [16,17,18]. In the case of domestic food safety, the importance of the risk perception as the background knowledge for the cause of improper behaviors of food handlers at home has been highlighted [3,4].

Hygienic handling practices based on proper knowledge and risk perception are essential to prevent diseases at the home [3,8,19]. However, many consumers have been improperly informed about the methods required to prevent foodborne disease at home [20]. International concern about food safety has prompted considerable studies to gain insights into domestic food-handling practices and risk perceptions mainly in the Western countries (including the United Kingdom, Northern Ireland, Canada, Germany, and Belgium) and the United States [2,8,21,22,23,24,25,26,27,28]. However, there has been only a limited number of studies in South Korea. In this study, we conducted a nationwide survey of primary food handlers to investigate consumers’ behaviors as well as risk perceptions at home in South Korea. Since food-handling behaviors and perceptions of consumers could be altered by trends over time, the surveys were conducted in both 2010 (N = 609; 1st survey will be called here “Year 2010”) and 2019 (N = 605; 2nd survey will be called here “Year 2019”) with the same questionnaires to track changes in behaviors and perceptions.

Although consumers’ perception has been revealed as the motivation behind high-risk behaviors associated with domestic food safety, a lack of empirical data demonstrating the unchanged perception-behavior gap acts as a major hurdle for conventional intervention strategies (e.g., advertising, providing guidelines, publishing pamphlets, providing education, etc.) to drive the alteration of the behaviors of the lay public. Since the previous research on consumers’ risk perception and/or high-risk behaviors regarding the domestic food safety have been conducted by the cross-sectional analyses [2,21,25,26,27,28], we expected that unchanged and/or emerging high-risk behaviors can be identified by the comparative analysis of two individual consumer surveys using sociodemographic characteristics of respondents on a decade basis. The major aim of this research is to identify the unchanged gap between the major risk perception and behaviors of consumers over time. The survey on the consumers’ practices linked to the risk perception was expected to identify the self-reported behaviors which can be regarded as the potential for the deviate behaviors [29,30,31]. In this study, we conducted a nationwide trend survey of primary food handlers (i.e., food consumers who are the main people involved in food preparation at home) to investigate consumers’ behaviors as well as risk perceptions at home in South Korea.

## 2. Materials and Methods

### 2.1. Questionnaires for Food-Handling Behaviors and Perception

To develop questionnaires, experts in different fields, representing the South Korean government (Ministry of Food and Drug Safety, Cheongju-si, Korea), food safety laboratories (Korea University, Seoul, Korea), consumer organizations, and a professional market research company (Gallup Korea, Seoul, Korea) formed a consulting committee. These experts compiled a draft questionnaire. The questions were determined based on the general food safety guidelines for the home provided by health authorities, including the US Food and Drug Administration (US FDA) and Food Safety and Inspection Service in the US Department of Agriculture [32,33]. Only general guidelines were included in a draft questionnaire while guidelines, not culturally applicable to Korean consumers, were excluded. A draft questionnaire was inspected with a consulting committee and revised to develop the final questionnaire.

A food safety perception questionnaire was designed to obtain information about food handlers’ perceptions about hazardous behaviors that frequently occurred at home. Detailed questions are shown in Table 1.

### 2.2. Pilot Test of Questionnaires

To confirm the clarity of the draft questionnaire a pilot test, using a draft questionnaire, was conducted with 15 randomly selected consumers and 15 expert researchers before performing the main survey to confirm the clarity of the draft questionnaire. The pretest consumers were asked to respond to the following questions: (1) Did you clearly understand the terminology used? (2) How much time did it take you to respond to all the questions? (3) Did the questionnaire contain unclear expressions? (4) Is there any question that is difficult to interpret? (5) Do any of the terms need to be clarified? (6) Did you feel displeasure or resistance when you responded to the questionnaire? (7) Did you have any opinions that differed from the existing response options? The questionnaires were revised based on the results of the pretest, specifically focusing on the understandability of questions. The answers of the pre-testers were not included in the survey results.

### 2.3. Surveys

This survey targeted adult consumers (>18 years old), mainly the primary food handlers in their homes, from South Korea. Before the survey, consumers (N = 609) were pre-allocated (i.e., quota sampling) according to population data from the statistical yearbook of South Korea using the multistage stratified systematic sampling method [34,35]. Participants of the survey were asked for their sociodemographic characteristics to obtain responses according to the pre-allocated population composition. A survey was conducted in households or shopping centers from various locations throughout Korea, including large cities (Busan, Daegu, Daejeon, Gwangju, Incheon, Seoul, and Ulsan), small and medium cities, and country towns. The sampling fraction used for the geographic location was proportional to the total population. All the respondents were interviewed face-to-face by trained panels (Gallup Korea). The instructions explaining the purpose of the study were displayed at the front of the questionnaire. The investigator briefly explained the purpose and nature of the present study. The questionnaires (a total of 24 questions) as described in Table 1 were used for both surveys. A survey in 2019 was conducted for targeted adult consumers (N = 605) using the same questionnaire and method for Year 2010. They were recruited by a multistage stratified systematic sampling method for the homogeneity of sociodemographic characteristics both within and between surveys to collect comparable responses. The average duration for the data collection process of both surveys was ca. 3 months for each time-point (2010 or 2019). As consumers’ responses for the risk perception and food handling practices in both surveys were obtained by same questionnaires organized as multiple-choice questions with a single-select answer option (except for the “Q10. How long do you store those foods in the selected storage place?” (an open question)), comparative analysis on the results from two time period (2010 and 2019) could be conducted. The sociodemographic characteristics of the participants (e.g., gender, age, location, level of education, number of family members, and average monthly income) are shown in Table 2. To evaluate the effect of the year on the survey results, Pearson chi-square tested the associations of each sample characteristic between the time-points of Year 2010 and Year 2019. There was no effect of the year for all variables (sociodemographic characteristics) considered in this study (i.e., gender, age (years), location, level of education, number of family members).

### 2.4. Data Analysis

All the questions and responses were manually coded by assigning a unique number with the sui generis data coding system used by Gallup Korea for each response, and the codes were entered into a multivariate Excel spreadsheet. As shown in Section 2.3, the Pearson chi-square test was conducted to evaluate the effect of the year to the survey results by the analysis of the association of each sociodemographic characteristic between Year 2010 and Year 2019. Kruskal–Wallis test method was used to evaluate the changes in common high-risk behaviors between surveys by the analysis of the significant differences (*p* < 0.05) of responses from Year 2010 and Year 2019. All statistical analyses (i.e., Pearson chi-square test, Kruskal–Wallis test) for the data were conducted by using the SPSS statistical package (Statistical Package for the Social Sciences, version 12.0, SPSS Inc. Chicago, IL, USA).

## 3. Results

### 3.1. Discordance Between Consumers’ Food Safety Perceptions and Behaviors (Risk Perception-Behavior Gap)

Figure 1, Figure 2 and Figure 3 showed that there were large gaps between consumers’ food safety perceptions and behaviors in surveys performed in 2010 and 2019, respectively. In general, a similar tendency was observed in both surveys, indicating that the gaps between safety knowledge and behaviors were not narrowed, even after 8 years. Detailed data for Figure 1, Figure 2 and Figure 3 with the results of statistical analysis were also indicated in Appendix A.

#### 3.1.1. Storage of Perishable Foods Without Washing and Trimming

As shown in Figure 1, many consumers (71.8% and 67.8% in the Year 2010 and Year 2019, respectively) perceived the storage of raw food materials (e.g., fruits, vegetables, meat, fish/shellfish) without any preparation, including washing/trimming, as hazardous. A considerable number of consumers (32.5%) did not wash or trim food before storage in Year 2010, and 33.7% of consumers did not wash perishables in Year 2019. Moreover, significant differences (*p* < 0.05) represented by the decreases in the response to proper behavior were observed: from 22.5% in Year 2010 to 7.9% in Year 2019 for “completely yes” (Figure 1); 51.7% in Year 2010 to 35.7% in Year 2019 for “always + frequently” (Appendix A). The response “moderate” (from 15.8% in Year 2010 to 30.6% in Year 2019) was also increased (*p* < 0.05).

#### 3.1.2. Thawing Foods at Room Temperature

Figure 2 showed the risk perception on thawing frozen foods at room temperature and practices with perspective to various thawing methods (e.g., using a refrigerator or microwave, placed at room temperature). Results of Year 2010 indicated that many consumers (63.2%) were knowledgeable about the hazard of thawing frozen foods at room temperature (mostly hazardous = 36.8%; completely hazardous = 26.4%). However, the interviewed consumers (53.5%) thawed frozen foods at room temperature, 36.9% on the countertop, and 16.6% at room temperature water. Only 42.1% of consumers properly thawed foods; 25.0 and 17.1% thawed them in the microwave oven and in the refrigerator for 1–2 days before use, respectively. In the case of Year 2019, 58.5% of the consumers thawed them at room temperature although approximately half of the consumers (48.1%) were knowledgeable about the relevant hazard. Especially responses to “Place on counter at room temperature” increased (*p* < 0.05) from 36.9% to 43.6%. Whereas significant decreases in the proper risk perception (i.e., the correct response in line with conventional food safety guidelines) in Year 2019 were noticeable, implying that the risk perception worsened between studies: from 26.4% to 17.4% (*p* < 0.05), from 36.8% to 30.7% (*p* < 0.05), and from 63.2% to 48.1% for the responses “completely hazardous”, “mostly hazardous”, and “completely hazardous + mostly hazardous”, respectively (Appendix A). The percentage of respondents who properly thawed food (using a microwave or placing foods in a refrigerator 1–2 days before use) was 38.5%.

#### 3.1.3. Improper Handling and the Storage of Leftovers

The gap between the risk perception and potential for deviant behavior regarding the leftovers was shown in Figure 3. In terms of Year 2010, although the respondents (64.0%) perceived the hazard of storing leftover food at room temperature, 95.6% of consumers (56.0% stored leftover foods in the refrigerator after chilling at room temperature, and 39.6% stored foods at room temperature) exposed leftovers to room temperature conditions. Only a few respondents properly handled leftovers: refrigerating them immediately (1.6%) or after chilling them in cold water (2.5%). Survey results in 2019 showed that 60.8% of the respondents perceived the hazard of storing leftovers at room temperature. While the increase in the response to “moderate” (from 23.3% to 32.4%) (*p* < 0.05) and the decrease in the response to “mostly safe” (from 11.2% to 6.3%) (*p* < 0.05) for risk perception resulted in the overall decrease in the proper risk perception (from 12.6% to 6.8% for the response to “completely safe + mostly safe”) (*p* < 0.05) (Appendix A). Most consumers (93.0%) exposed leftovers to room temperature conditions, and only a small percentage of respondents properly handled leftovers, including placing them in a refrigerator after chilling in cold water (3.5%) and immediately placing them in a refrigerator (3.5%). Distinct increases in the responses to “store in the refrigerator after chilling at room temperature” were analyzed between survey rounds (from 56.0% to 68.9%; *p* < 0.05) despite the decreases in the responses to ”store at room temperature” between survey rounds (from 39.6% to 24.1%; *p* < 0.05).

### 3.2. Changes in Common High-Risk Behaviors Between Surveys

Other common high-risk behaviors of consumers in 2010 were summarized in Table 3: (1) Packaging and storage of foods, (2) Management of cutting board, (3) Management of kitchen cloth.

Common risk food handling practices of consumers could be identified by the analysis of results from Year 2010. When purchased foods are too large to cook or eat, cutting or dividing them into smaller pieces is effective for handling and storage. Most consumers (mostly yes, 35.5%; completely yes, 34.5%) responded that they divided large portion of foods into smaller pieces for storage; however, only 12.5% (mostly yes, 8.9%; completely yes, 3.6%) of the consumers wrote information (e.g., product name, shelf life, proper storage method) on the divided foods). Therefore, a considerable number of consumers could not obtain information about divided foods after storage, which could lead to the mishandling of foods. When consumers were asked how long they used cutting boards, only 2.8% of consumers used cutting boards for < 1 year, while 69.3% reported using them for > 3 years. In terms of the question for the sanitization of kitchen utensils, 68.0% of respondents sanitized the cutting board. In the case of kitchen cloths, respondents usually replaced them once for every 3 months (33.7%), 1–2 times in a month (31.9%), and once or more in a week (19.4%). The percentage of respondents who sanitized kitchen cloths was 87.7%.

According to the results of Year 2019, common high-risk behaviors in Year 2010 were improved after 9 years (Table 3). Although both the increases (*p* < 0.05) and decreases (*p* < 0.05) in the negative responses (i.e., Never + Seldom) and the positive responses (i.e., Always + Frequently) were observed, respectively, more than half of consumers (57.9%) responded that they divided the large portion of foods into smaller portions. Although a substantial percentage (48.3%) of respondents did not write information, more consumers (25.6%) compared to Year 2010 (*p* < 0.05) indicated the information on divided foods. Over time, the period of using cutting board was shortened; distinct increases (*p* < 0.05) in the responses to 1–2 years (30.6%) and decreases (*p* < 0.05) in > 3 years (41.3%). The usage period of kitchen cloth was also shortened in Year 2019 compared to Year 2010; more consumers (*p* < 0.05) replaced it once or more than in a week (26.6%), and fewer consumers (*p* < 0.05) replaced it once in 3 months (25.3%). The percentages of respondents who sanitized cutting board and kitchen cloth decreased (from 68.0% to 44.8% and from 87.7% to 59.3%, respectively; *p* < 0.05), which was likely due to the frequent replacement of kitchen utensils rather than sanitizing them.

### 3.3. Proper Behaviors of Consumers (Reported to Practice Proper Food Handling in Line with Guidance)

Figure 4, Figure 5 and Figure 6 showed that consumers practiced proper behaviors in accordance with their perceptions in both surveys (Detailed data with the results of statistical analysis were also indicated in Appendix A).

#### 3.3.1. Risk Perception and Behaviors of Consumers

The majority of consumers (83.4% for Year 2010, 77.8% for Year 2019) perceived the hazard of touching cooked foods after handling raw material (meat, fish/shellfish, eggs) without washing hands, and substantial consumers (74.3% for Year 2010, 67.1% for Year 2019) washed hands after handling potentially hazardous foods, reducing the risk of cross-contamination (Figure 4). Although there were decreases in the responses to proper risk perceptions (from 83.4% in Year 2010 to 77.9% in Year 2019 for the response “completely hazardous + mostly hazardous”) and hygienic behaviors (from 74.2% in Year 2010 to 67.1% in Year 2019 for the response “always + frequently”), distinct trends that consumers confirmed the food safety rule with proper risk perception were maintained in both Year 2010 and Year 2019. Increases in the responses as “moderate” were also observed in both risk perception (from 13.3% to 17.4%; *p* < 0.05) and behavior (from 8.7% to 17.5%; *p* < 0.05) (Figure 4).

Of the respondents, most participants perceived the hazard of not fully cooking foods (90.0% and 80.0% in Year 2010 and Year 2019, respectively) with their self-reported behaviors supporting those risk perceptions (94.1% and 85.6% of consumers fully cook foods in Year 2010 and Year 2019, respectively) (Figure 5).

Generally, consumers reheated leftovers before eating them (91.9% for Year 2010, 85.0% for Year 2019), and most consumers (76.8% for Year 2010, 66.8% for Year 2019) were knowledgeable that eating leftovers without reheating was hazardous (Figure 6). Responses from Year 2010 to Year 2019 regarding the proper risk perception ((from 50.9% to 43.5% for ”mostly hazardous” (*p* < 0.05), from 76.8% to 66.8% for ”completely hazardous + mostly hazardous”) (*p* < 0.05)) and proper behavior (from 45.6% to 34.4% for ”frequently” (*p* < 0.05), from 92.0% to 85.0% for “always + frequently” (*p* < 0.05)) decreased by the increases in the responses ”moderate” for both the risk perception (from 17.4% to 26.6%; *p* < 0.05) and behavior (from 6.7% to 12.2%; *p* < 0.05) of consumers (Appendix A).

#### 3.3.2. Storage Places/Periods for Food

Table 4 showed the storage places/periods for food in 2010 and 2019, respectively. Food safety authorities recommended that fresh eggs should be stored in refrigerators for 3–5 weeks [36], and consumers usually kept them in refrigerators (96.9 and 90.6% in Year 2010 and Year 2019, respectively) for fewer days than recommend (11.7 and 14.5 days on average, respectively). The US FDA (2018) also proposed the proper storage length and location for meat, chicken, fish, and shellfish as follows: 3–5 days in the refrigerator and 6–12 months in the freezer for meat (steak), 1–2 days in the refrigerator and 1 year in the freezer for chicken (whole chicken or turkey), 1–2 days in the refrigerator and 6-8 months in the freezer for fish (lean fish), and 1–2 days in the refrigerator and 3–6 months in the freezer for shrimp. Most consumers generally followed the rules for food storage. In terms of the storage places, significant differences (*p* < 0.05) in the responses on the preferred place for each food item between Year 2010 and Year 2019 were mainly observed (except for refrigerator and freezer for chicken and fish; other environments at room temperature for meat, shellfish, and milk; do not know/no response for all food items), but distinct trends for the preference were maintained as follows: meat for freezer, chicken for refrigerator or freezer, fish for freezer, shellfish for freezer, fruit and vegetables for refrigerator, eggs for refrigerator, milk for refrigerator, and frozen processed foods for freezer.

#### 3.3.3. Other Behaviors of Consumers

Other consumers’ behaviors were also shown in Table 5. In short, the most consumers properly followed the food safety rules (at least 70% of the consumers): (1) store foods separately (thaw the only portion of frozen foodstuffs for cooking, store remaining raw foodstuffs with sealing after cooking), (2) store foods in the proper place and for the proper period, (3) wash fruits and vegetables properly before eating, (4) wash hands after handling potentially hazardous foods, (5) cook food to eat, and (6) reheat leftovers to eat. Although some answer options showed changes over time (i.e., a significant decrease of increase of the responses (*p* < 0.05)), distinct differences between the answer options within the results from each survey were obviously observed.

## 4. Discussion

In the present study, we conducted multiple individual surveys for the demonstration of the distinct/obvious gap between risk perception and practices regardless of the time-point for the surveys. This novel approach can contribute to the body of the knowledge on consumers’ risk perception-behaviors by overcoming a limitation of relevant research which have been conducted as the singular cross-sectional study for specific time-point [2,21,25,26,27,28]. Comparative analysis of multiple individual surveys conducted in different time-points was rarely reported from the research not only for consumers [1,21,37,38,39,40,41,42,43] but also for food handlers in food services [44,45,46,47,48,49,50,51]. This is the first study regarding the comparative analysis of the surveys from multiple time-points to identify the distinct/obvious gap between consumers’ risk perception and practices which were not improved over time. Previous consumer surveys regarding the domestic food safety identify the risk perception and behaviors in the specific time-point [1,21,37,38,39,40,41,42,43], however, the results from research cannot demonstrate whether the risk perception and/or behaviors will be changed over time. This limitation implies that the risk perception-behavior gap described by the survey conducted in a singular specific time-point cannot represent the general responses due to the time-dependent variability of consumers’ perception and behaviors. Whereas most previous research have mainly conducted not only in the specific time-point but also from the specific region as follows: Belgium [41], China [40], Island of Ireland [21], Poland and Thailand [39], Republic of Ireland [37], Slovenia [1], Trinidad [43], Turkey [38], and USA [42]. Although previous research for the survey of respondents from multiple nations have been reported [39,48], comparative analysis between the results of research from other countries was rarely conducted. Since the risk perception and/or behaviors are different according to the regions and changeable over time, both regions and time-points can act as determinant factors for the survey results. Thus, the comparative analysis among the survey results with the perspectives to the region (i.e., countries) was rarely conducted even for the integrative reviews regarding the consumer surveys on food safety [4,52]. While the impact of the regional factor can be estimated by the comparative analysis of research which adopted the multiple individual surveys for general consumers (i.e., respondents selected by considering the homogeneity of sociodemographic characteristics) from various time-points because the longitudinal study can support the generalizability of the results with the perspectives to the time as a determinant factor. Consequently, further survey research based on our study design with the consideration of the time-points in other countries are expected to identify the most important food handling deficiencies by the analysis of the unchanged risk perception-behaviors over time regardless of the nations. Moreover, the accumulation of the survey data feasible to the comparative analysis between relevant research can also be useful for the modeling among the knowledge-attitude-practice regarding the food safety to predict the level of the risks and to establish the risk management strategies [53,54].

This study attempted to provide empirical data about food-handling behaviors as well as perceptions of food safety at home through consumer surveys focused on primary food handlers in 2010 and 2019. From the present study, we could follow the major trends for food safety knowledge and provide practical information about the high risk or proper behaviors implemented at home. Major practices properly implemented in accordance with consumers’ perceptions and high-risk behaviors that did not support consumers’ confidence in food safety were identified. Although the improvement in high-risk behaviors over a decade (from Year 2010 to Year 2019) and proper behaviors observed in both surveys were also noticed, several common high-risk behaviors were not corrected. The unchanged gap between risk perception and practices of consumers should be considered as the endemic problems for domestic food safety and the blind spots of the current intervention strategies against high-risk behaviors which required novel countermeasures. There were large gaps of perception-behaviors in 2010, including (1) storing perishable foods without any preparation (washing or trimming), (2) thawing foods at room temperature, and (3) exposing leftovers to danger zone temperatures. These gaps between consumer perceptions and behaviors in 2010 remained in 2019.

Since the internationally-recognized consumer guidelines suggested cleaning as a basic step to food safety at home [32], previous research regarding the washing perishables have consistently reported the proper perceptions with actual practices [39,40] whereas there was a lack of survey research washing raw materials prior to storage. However, in this study, the storage of perishable foods without washing and trimming was analyzed as a major potential for deviant behaviors in both surveys despite consumers’ proper safety perception/knowledge. Preparation of perishables before storage can prevent not only the risk of foodborne diseases by the removal of contaminants attached to foods but also the cross-contamination during the storage of various foodstuffs [55,56]. These contributions to food safety should be emphasized by the educational tools (e.g., guidelines, programs, leaflets) regarding the preparation followed by the storage of perishables. The importance for washing and trimming of foods mainly for eaten as raw (e.g., fruits, vegetables, etc.) has been highlighted by the previous research on the effectiveness of the decontamination of microbiological risk factors (i.e., pathogens) [57,58] and/or the validation of the cross-contamination [59]. Moreover, the establishment and the improvement of the knowledge basis regarding proper packaging methods (e.g., suggestions on how not to group a large amount of food together, how to separate large pieces in individual packages as an alternative to cutting them into smaller pieces, and how to place them properly in the freezer to ensure optimum quick freezing, etc.) should also be followed to alter the consumers’ high-risk behaviors.

Frozen foods should not be thawed at room temperature to avoid the exposure of foods to “danger zone” (4.4–60.0 °C; 40–140 °F), the temperature range that can cause the growth of foodborne bacteria in foods [60,61]. Thawing frozen foodstuffs at room temperature have been reported as general behaviors or underrecognized risk perceptions of consumers in other relevant research [21,41,43]. Thawing in a refrigerator or in cold water with the time-temperature control has not been generally preferred than the exposure to room temperate [43,62], and other inadequate methods using tap water [63] or immersing in warm water [64] were also frequently used. Whereas a direct gap of perceptions-behaviors observed in this study has been rarely highlighted by previous studies. Low-risk perception on the temperature control has been suggested as the major cause for the high-risk behaviors of thawing frozen foods from relevant studies [39,40], however, this research implied that high-risk behavior could be occurred by consumers with proper risk perception.

Leaving food out at an improper temperature is one of the main factors commonly associated with foodborne disease at home [3]. The government recommended the following practice for leftovers: to keep food out of the “danger zone”, wrap leftovers well, store leftovers safely, thaw frozen leftovers safely, reheat leftovers without thawing, and reheat leftovers [65]. Leftovers should be refrigerated or frozen quickly as soon as possible to prevent the exposure of foods under favorable conditions for microbial growth [1]. However, in this research, exposure leftovers at room temperature have been reported as one of the common high-risk behaviors of primary food handlers. This risk factor has been also highlighted in recent studies on consumer surveys which reported the preference for cooling leftovers at room temperature before the storage in the refrigerator [54,62,66].

Whereas the improvement on consumers’ high-risk behaviors (Section 3.2) or proper behaviors observed in both surveys (Section 3.3) can lower the microbiological risks of foodborne diseases, however, the importance of overall procedures for consumers’ food handling should not be underestimated. Investigations for the identification of underreported potential risks in domestic food safety linked to the consumers’ perceptions and behaviors should be persistently conducted. In the case of handling kitchen utensils, although increases in proper practices for the management of the cutting board and kitchen cloths were observed in Year 2019 (Table 3), improper behaviors of consumers for the use of those kitchen utensils to prepare foods could result in the foodborne illnesses despite the hygienic management because cross-contamination can occur when harmful bacteria were transferred to food from kitchen utensils [67]. Especially reusing the same utensils for various kinds of foods including raw materials and the ready-to-eat products has been reported as representative practices of poor sanitation procedures which could result in cross-contamination [43]. Moreover, previous studies on consumer surveys for domestic food safety also highlighted various other cross-contamination routes from raw materials to cooked foods (e.g., via knives, cutting boards, and/or plates, etc.), highlighting the necessities on the systematic intervention structures for kitchen utensils [41]. As both risk perception and behaviors regarding the separated use of kitchen utensils is regarded as a major risk factor [1,40], internationally-recognized consumer guidelines have suggested “separate” as keywords for food safety in the kitchen [32,68]. A recent report reported that most of the respondents from the survey on food handlers’ behaviors with the perspective of the meal preparation at home declared that they mainly separated kitchen utensils including the cutting boards for raw and cooked foods [54]. As shown in this study, a significant number of respondents from the relevant previous research also reported consumer preference for washing utensils in hot water with detergent to cut various kinds of raw materials by using the same cutting board [21,66]. Thus, the channels and methods of the delivery of proper information for the effective conditions to eliminate the major microbiological factors present on utensils should be established [69,70] because of the incomplete sterilization of cutting board by using hot water and detergents has been reported [71,72].

The results obtained from 2010 and 2019 suggested that behaviors did not support consumers’ confidence in food safety. These gaps between perceptions and behaviors might be significant risk factors for foodborne disease and could increase the likelihood of food deterioration/poisoning. The result of this study highlights that education tools for improving risk perceptions and knowledge should have been implemented [73,74,75] because these high-risk behaviors are generally based on the lack of awareness of domestic food safety. The intervention for the improvement of consumers’ poor food safety knowledge and/or perception-behaviors reported from most previous research has been also limited to education [1,21,38,39,40,42,43]. However, since there is no enforceable regulation for food handling at home, consumers can easily ignore the importance of handling practices for food safety [11]; this can be a key reason why the gaps were not narrowed over a decade in this research. Thus, strategies for the improvement of the effectiveness and efficiencies of the education should also be adopted (e.g., highlighting responsibility for food safety as a primary food handler, informing the susceptibility and the severity of outcomes, building confidence, etc.) [4]. Moreover, the information sources and/or channels have been reported as the significant determinant factors for the educational effects to consumers [38,76,77], highlighting various education tools which have been regarded as effective strategies for behavior intervention and/or information delivery should also be considered (e.g., the social media and web-based communication with consumers) [52,78,79,80]. To narrow a perception-behavior gap, education should focus on consumers who think of themselves as knowledgeable regarding food safety issues, and communication with those consumers to recognize the gap is expected to contribute to changes in their hygienic practices [81,82,83,84]. As a moderate response to the questions of risk perception-behaviors can also be regarded as the potential for behavior that deviates from the best practice guidelines, further study regarding the in-depth examination on the intention of consumers who responded with the “moderate responses” is expected to change moderate to proper responses [29,30,31]. Moreover, further studies based on the segmentation analysis of consumers are needed to identify the major education targets that demonstrate high levels of knowledge regarding best practice but tendencies to report deviant behaviors and those with low knowledge and therefore likelihood to also demonstrate deviant behaviors [85]. A continuous education program from health authorities on the potential hazards of improper food handling is essential to motivate changes to improper practices at home using a varied approach including practical guidelines, face-to-face education programs, and the distribution of written materials such as leaflets. We also suggest that future food safety management should be further specialized for the perception-practices gap and proper behavior by the behavioral interventions. Even though most consumer studies have been mainly limited to the surveys investigating current status without the application of the interventions [45], findings from the research on the behavior intervention methods for food handlers in food services (e.g., training program, the legislation of policy and/or regulations for food safety, the inspection, the supervision, etc.) [45,46,47,49,50] can be applied to support the improvement on the educational tools and/or programs for consumers. Advanced strategies for consumer education are expected to develop consumers’ recognition of food safety issues and reduce foodborne illness at home.

## 5. Conclusions

The results of the present study provided comprehensive data about behaviors and perceptions of consumers (particularly primary food handlers at home) and could be used as an information basis for the development of educational tools specified for food consumers. Whereas the limitations for the study design implied the necessity of further research as follows: (1) the consideration of the recent changes in the increases of male food handlers, (2) the analysis of the responses from the same respondents in multiple individual surveys, and (3) in-depth examination of the determinant factors which can induce the changes in risk perception and/or behaviors and the evaluation on the specific intervention against the risk perception-behavior gap (e.g., advertising, providing guidelines, publishing pamphlets, providing education, etc.). Firstly, we unavoidably used a heavily female-biased sample because of the distinct distribution of male and female primary food handlers at home in South Korea, so the further survey with the increases in the distribution of male food handlers in South Korea is likely to identify emerging high-risk behaviors from males. Secondly, direct comparison for the same respondents and two time periods can effectively show the impact of time in the high-risk behaviors and perception of consumers. Whereas as this research aimed to obtain general responses from consumers (i.e., representative sample) in each survey, we recruited the respondents not only from Year 2010 but also Year 2019 to use a representative sample in each time-point of the survey. Thirdly, as this research did not focus on the specific intervention strategy for domestic food safety and determinant factors which might affect consumers’ risk perception and/or behaviors, the further examination on the effectiveness of various interventions is expected to find out optimal methods against risk perception-behavior gap observed in this study. In conclusion, our findings supported the understanding of the risks in domestic food safety necessary for the development of effective perception-behavior interventions to narrow the risk perception-behavior gap. This study is expected to act as a leading role of the representative work for the novel research design (i.e., comparative analysis on the individual consumer surveys conducted with same questionnaires at different time-points for identifying unchanged distinct gaps in risk perception-behaviors over time), highlighting the necessity for the following surveys from various regions and time-points to expand the body of knowledge on food safety for consumers.

## Figures and Tables

**Figure 1 foods-09-01457-f001:**
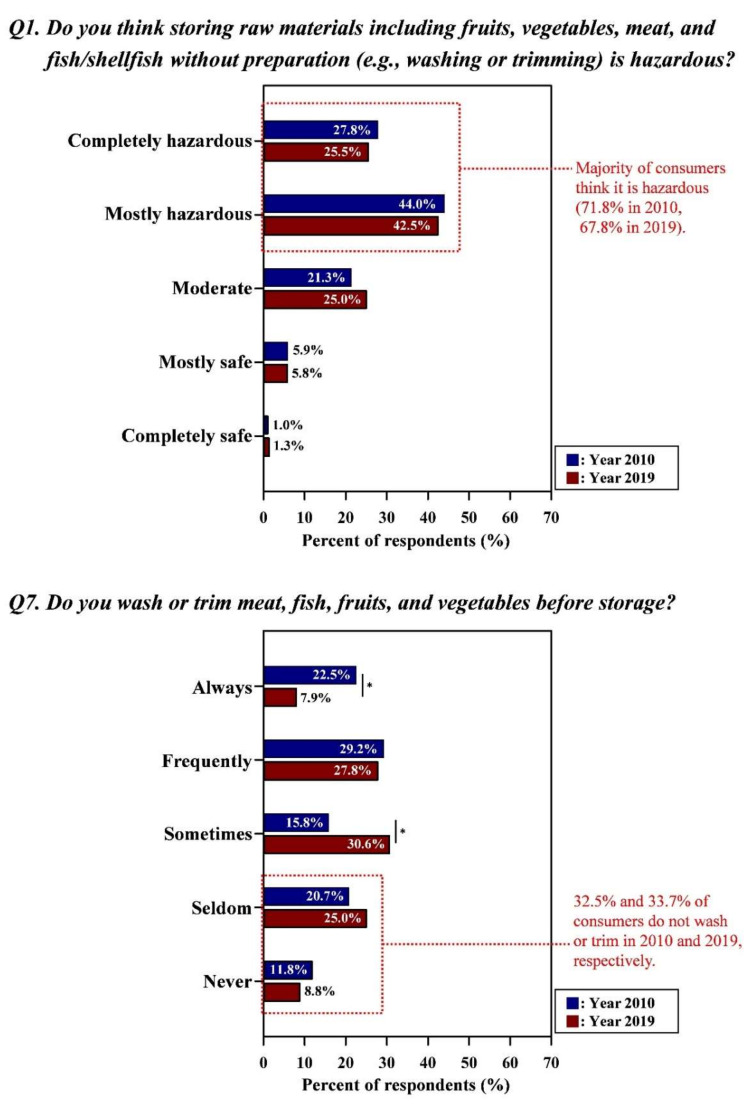
Gap between the consumers’ perceptions and behaviors regarding the storage of perishable foods without washing and trimming. Asterisk (*) indicated between graphs means the significant differences of responses (the Kruskal-Wallis test; *p* < 0.05) in each answer option from Year 2010 and Year.

**Figure 2 foods-09-01457-f002:**
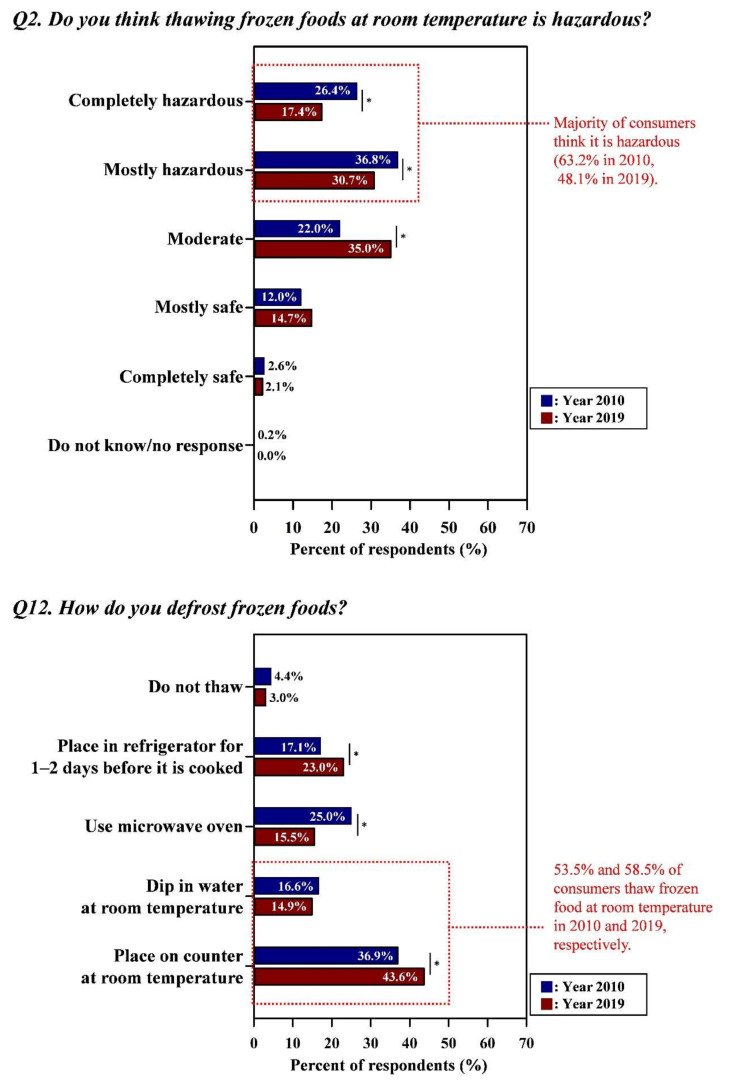
The gap between the consumers’ perceptions and behaviors regarding frozen foods. Asterisk (*) indicated between graphs means the significant differences of responses (the Kruskal–Wallis test; *p* < 0.05) in each answer option from Year 2010 and Year 2019. Red letters indicated the improper risk perceptions (i.e., the response in opposition to conventional food safety guidelines) and high-risk behaviors of consumers. All data regarding the percentage of the combined answer options (e.g., Always + Frequently) are provided in Appendix A.

**Figure 3 foods-09-01457-f003:**
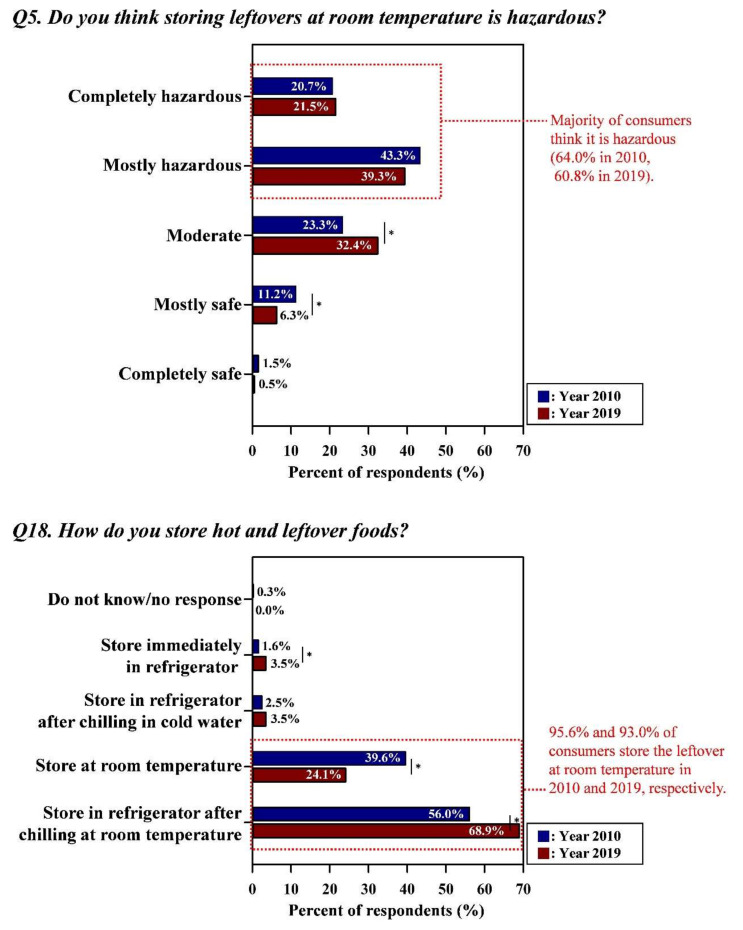
The gap between the consumers’ perceptions and behaviors regarding the leftovers. Asterisk (*) indicated between graphs means the significant differences of responses (the Kruskal–Wallis test; *p* < 0.05) in each answer option from Year 2010 and Year 2019. Red letters indicated the improper risk perceptions (i.e., the response in opposition to conventional food safety guidelines) and high-risk behaviors of consumers. All data regarding the percentage of the combined answer options (e.g., Always + Frequently) are provided in Appendix A.

**Figure 4 foods-09-01457-f004:**
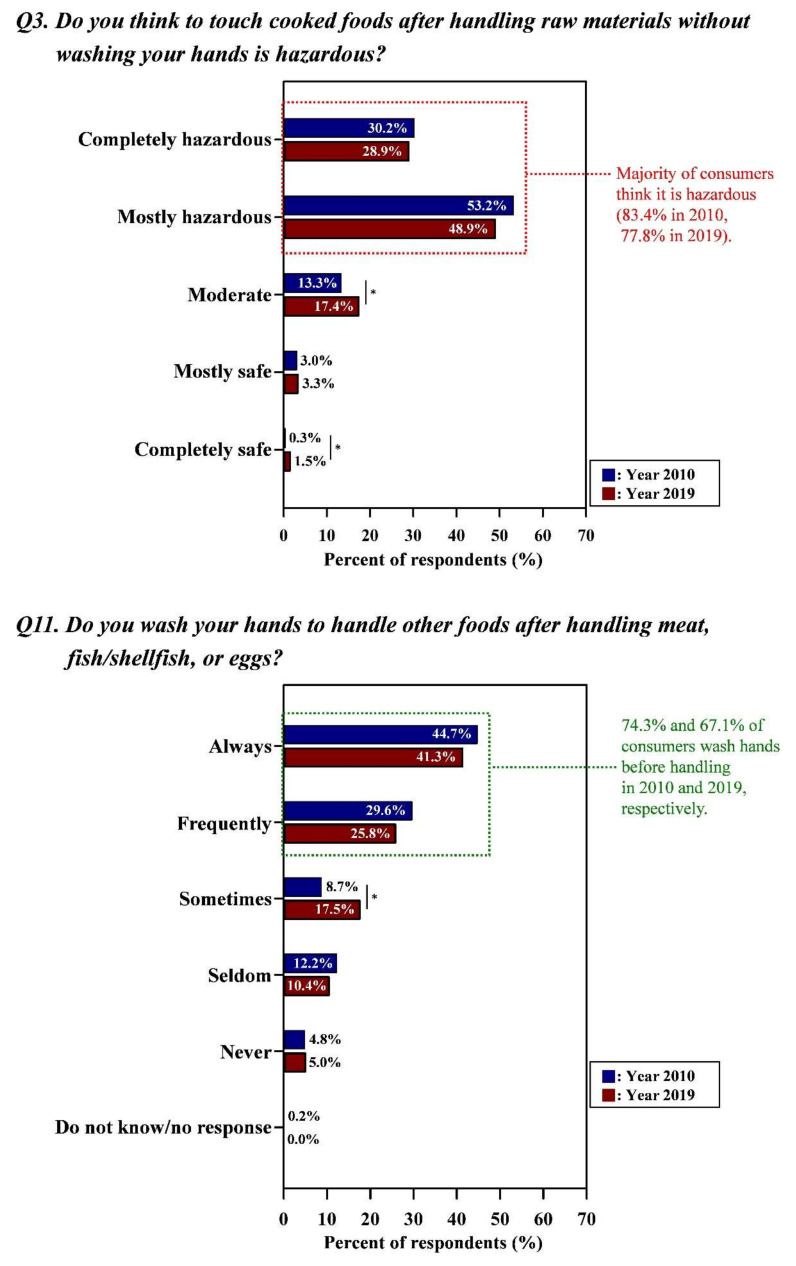
Proper behaviors of consumers in accordance with their perceptions regarding the washing hands. Asterisk (*) indicated between graphs means the significant differences of responses (the Kruskal–Wallis test; *p* < 0.05) in each answer option from Year 2010 and Year 2019. Green letters indicated the proper risk perceptions (i.e., the correct response in line with conventional food safety guidelines) and behaviors of consumers. All data regarding the percentage of the combined answer options (e.g., Always + Frequently) are provided in Appendix A.

**Figure 5 foods-09-01457-f005:**
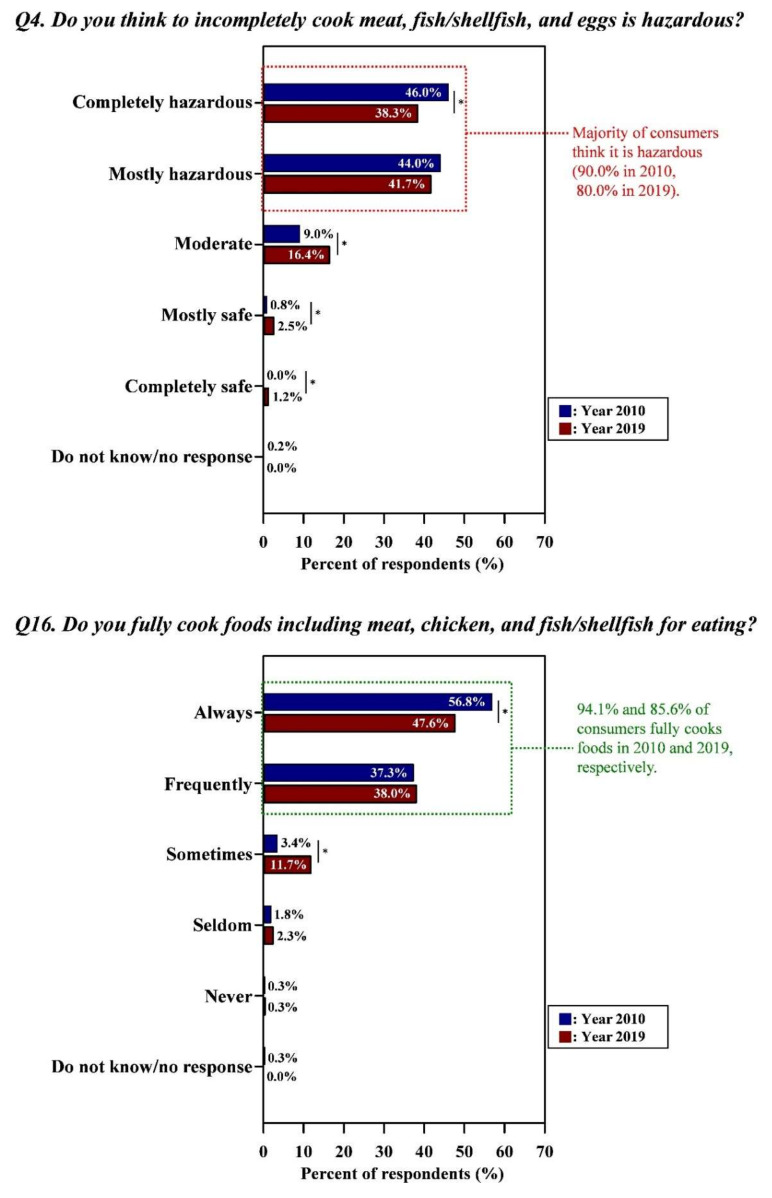
Proper behaviors of consumers in accordance with their perceptions regarding incomplete cooking. Asterisk (*) indicated between graphs means the significant differences of responses (the Kruskal–Wallis test; *p* < 0.05) in each answer option from Year 2010 and Year 2019. Green letters indicated the proper risk perceptions (i.e., the correct response in line with conventional food safety guidelines) and behaviors of consumers. All data regarding the percentage of the combined answer options (e.g., Always + Frequently) are provided in Appendix A.

**Figure 6 foods-09-01457-f006:**
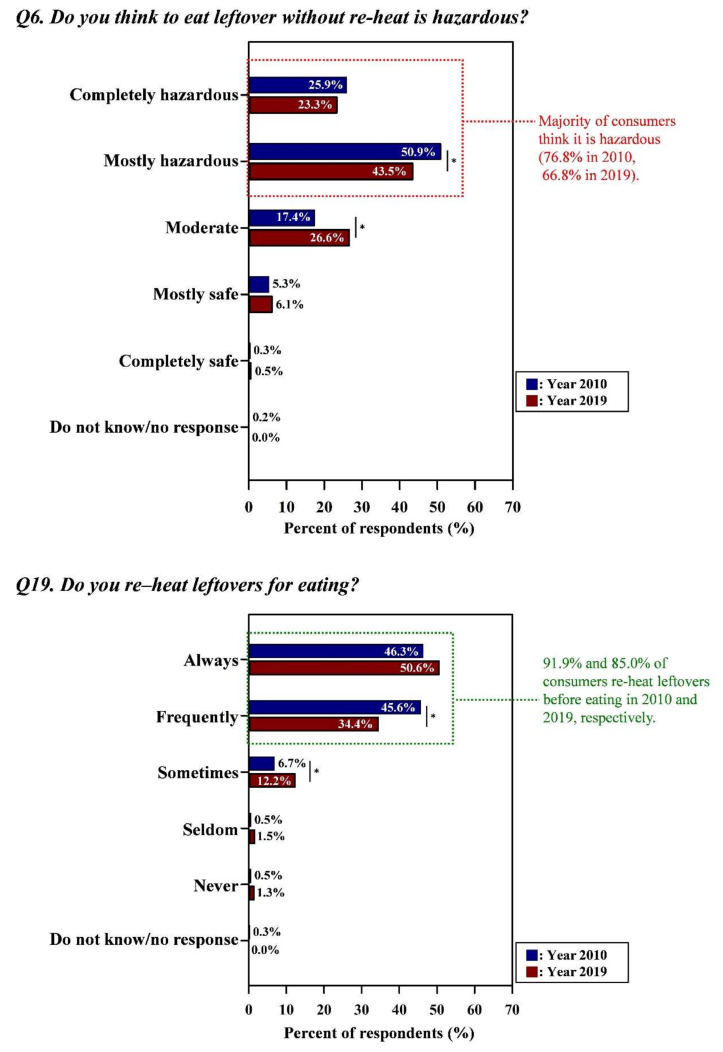
Proper behaviors of consumers in accordance with their perceptions regarding reheating leftovers. Asterisk (*) indicated between graphs means the significant differences of responses (the Kruskal–Wallis test; *p* < 0.05) in each answer option from Year 2010 and Year 2019. Green letters indicated the proper risk perceptions (i.e., the correct response in line with conventional food safety guidelines) and behaviors of consumers. All data regarding the percentage of the combined answer options (e.g., Always + Frequently) are provided in Appendix A.

**Table 1 foods-09-01457-t001:** Composition of the questionnaire.

Topic	Questions ^1^	Answer Options ^2^
Consumer’s perception	Q1. Do you think storing raw materials including fruits, vegetables, meat, and fish/shellfish without preparation (e.g., washing or trimming) is hazardous?	Completely hazardous Mostly hazardous Moderate Mostly safe Completely safe
	Q2. Do you think thawing frozen foods at room temperature is hazardous?	
	Q3. Do you think to touch cooked foods after handling raw materials without washing your hands is hazardous?	
	Q4. Do you think to incompletely cook meat, fish/shellfish, and eggs is hazardous?	
	Q5. Do you think storing leftovers at room temperature is hazardous?	
	Q6. Do you think to eat leftover without re-heat is hazardous?	
Food storage behavior	Q7. Do you wash or trim meat, fish, fruits, and vegetables before storage?	Always Frequently Sometimes Seldom Never
	Q8. When you buy a large portion of food, do you divide it into small portions for storage?	
	Q9. When you store a large portion of food in smaller portions, do you label the small portions with information about those foods?	
	Q10. Where do you store foods (meat, chicken, fish/shellfish, fruit/vegetables, eggs, milk, and frozen processed foods) among refrigerator, freezer, or other environments at room temperature? Then, how long do you store those foods in the selected storage place? ^3^	<Answer options for the storage place> Refrigerator Freezer Other environments at room temperature Do not purchase <The storage period was asked as an open question>
Preparing and cooking behavior	Q11. Do you wash your hands to handle other foods after handling meat, fish/shellfish, or eggs?	Always Frequently Sometimes Seldom Never
	Q12. How do you defrost frozen foods?	Do not thaw Place in the refrigerator for 1–2 days before it is cooked Use a microwave oven Dip in the water at room temperature Place on the counter at room temperature
	Q13. Do you thaw only a portion of food as much as you intend to cook rather than thawing all of the foods?	Always Frequently Sometimes Seldom Never
	Q14. Do you use plastic gloves when you handle meat, fish/shellfish, or eggs?	
	Q15. What do you do when you have a slight wound on your hand during cooking?	Keep cooking after the wound has been treated Keep cooking without any treatment Do not cook
Eating behavior	Q16. Do you fully cook foods including meat, chicken, and fish/shellfish for eating?	Always Frequently Sometimes Seldom Never
	Q17. Do you wash fruits and vegetables before eating?	
Management of leftovers	Q18. How do you store hot and leftover foods?	Store immediately in the refrigerator Store in refrigerator after chilling in cold water Store at room temperature Store in refrigerator after chilling at room temperature
	Q19. Do you re–heat leftovers for eating?	Always Frequently Sometimes Seldom Never
	Q20. Do you store remaining raw foodstuffs with sealing after cooking?	
Management of domestic kitchen utensils	Q21. How long do you use a cutting board?	Less than 6 months 6 months–1 year 1–2 years 2–3 years More than 3 years Other
	Q22. Do you sanitize your cutting board?	Yes No
	Q23. How often do you replace your kitchen cloth?	1 or more than in a week 1–2 times in a month Once in 3 months Once in 6 months Once in a year or more than a year Do not replace
	Q24. Do you sanitize your kitchen cloth?	Yes No

^1^ All questions except for the “Q10. How long do you store those foods in the selected storage place?” (an open question) were in multiple-choice questions with a single-select answer option. All answer options (i.e., choices) for each question are provided in Tables and Figures of this study. ^2^ All questions include the “Do not know/no response” as an answer option (this option is omitted in this Table to avoid the repetition). ^3^ Respondents were asked to select the storage place (refrigerator, freezer, or other environments at room temperature) for each food (meat, chicken, fish/shellfish, fruit/vegetables, eggs, milk, and frozen processed foods) and to answer the storage period as a unit of “days” for each food in the selected storage place.

**Table 2 foods-09-01457-t002:** Sociodemographic characteristics of the respondents and chi-square test results for each variable.

Variables	Number of Respondents to Year 2010 (*n* = 609) ^1^	Number of Respondents to Year 2019 (*n* = 605) ^2^	χ^2^	*p*-Value
Gender			0.005	0.943
Male	49	48		
Female	560	557		
Age (years)			0.604	0.963
19–29	67	74		
30–39	144	144		
40–49	148	145		
50–59	112	112		
>60	138	130		
Location			0.003	0.998
Large city	270	269		
Small or medium city	271	269		
Country town	68	67		
Level of education			2.003	0.572
Less than high school	122	120		
High school	231	231		
University	254	254		
No response	2	-		
Number of family members			3.620	0.306
One person	64	47		
2–3 persons	246	267		
4–5 persons	266	257		
More than 6 persons	33	34		

^1^ 1st survey conducted in 2010 will be called here “Year 2010”. ^2^ 2nd survey conducted in 2019 will be called here “Year 2019”.

**Table 3 foods-09-01457-t003:** The common high-risk behaviors of consumers were observed from surveys performed in 2010 and 2019.

Topic	Questions and Choices	Percent of Respondents or Answers		p-Value ^1^
		Year 2010 (N = 609)	Year 2019 (N = 605)	
Packaging and storage of foods	Q8. When you buy a large portion of food, do you divide it into small portions for storage?			
	Never	5.1%	5.8%	0.594
	Seldom	11.8%	16.0%	0.034
	Sometimes	13.1%	20.3%	0.001
	Frequently	35.5%	40.0%	0.103
	Always	34.5%	17.9%	0.000
	<Analysis of the combined responses> ^2^			
	Never + Seldom	16.9%	21.8%	0.031
	Always + Frequently	70.0%	57.9%	0.000
	Q9. When you store a large portion of food in smaller portions, do you label the small portions with information about those foods?			
	Never	36.1%	17.0%	0.000
	Seldom	39.1%	31.2%	0.000
	Sometimes	12.3%	26.1%	0.000
	Frequently	8.9%	20.7%	0.000
	Always	3.6%	5.0%	0.247
	<Analysis of the combined responses> ^2^			
	Never + Seldom	75.2%	48.3%	0.000
	Always + Frequently	12.5%	25.6%	0.000
Management of cutting board	Q21. How long do you use a cutting board?			
	Less than 6 months	0.5%	3.8%	0.000
	6 months–1 year	2.3%	6.1%	0.001
	1–2 years	12.0%	30.6%	0.000
	2–3 years	13.6%	18.2%	0.03
	More than 3 years	69.3%	41.3%	0.000
	Other	2.3%	0.0%	0.000
	Q22. Do you sanitize your cutting board?			
	Yes	68.0%	44.8%	0.000
	No	32.0%	55.2%	0.000
Management of kitchen cloth	Q23. How often do you replace your kitchen cloth?			
	1 or more than in a week	19.4%	26.6%	0.003
	1–2 times in a month	31.9%	35.9%	0.140
	Once in 3 months	33.7%	25.3%	0.001
	Once in 6 months	10.7%	8.1%	0.124
	Once in a year or more than a year	4.3%	4.1%	0.905
	Do not replace	0.2%	0.0%	0.319
	Q24. Do you sanitize your kitchen cloth?			
	Yes	87.7%	59.3%	0.000
	No	12.3%	40.7%	0.000

^1^*p*-value was provided according to the results of the Kruskal–Wallis test conducted for the analysis on the differences of responses in each answer option between Year 2010 and Year 2019. ^2^ To identify the difference between positive responses (i.e., Always + Frequently) and negative responses (i.e., Never + Seldom), answer options were combined.

**Table 4 foods-09-01457-t004:** The storage place and periods for specific foods in Year 2010 and Year 2019.

Q10. Where do you store foods (meat, chicken, fish/shellfish, fruit/vegetables, eggs, milk, and frozen processed foods) among refrigerator, freezer, or other environments at room temperature? Then, how long do you store those foods in the selected storage place?															
	**Refrigerator**			**Freezer**			**Other Environments at Room Temperature**			**Do Not Know** **/No Response**			**Do Not Purchase**		
	**Year 2010**	**Year 2019**	***p*-Value ^1^**	**Year 2010**	**Year 2019**	***p*-Value**	**Year 2010**	**Year 2019**	***p*-Value**	**Year 2010**	**Year 2019**	***p*-Value**	**Year 2010**	**Year 2019**	***p*-Value**
Meat (pork and beef)			0.000			0.000			0.988		-	0.158	-	-	-
Storage place	20.9%	30.3%		77.0%	67.9%		1.8%	1.8%		0.3%					
(storage period; day)	(2.5)	(3.8)		(11.6)	(14.2)		(1.3)	(4.7)		(4.5)					
Chicken			0.253			0.492			0.000		-	0.158	-	-	-
Storage place	44.8%	48.1%		48.3%	50.2%		6.6%	1.7%		0.3%					
(storage period; day)	(1.7)	(3.0)		(10.0)	(12.3)		(1.4)	(2.9)		(4.5)					
Fish			0.901			0.893			0.033		-	0.158	-	-	-
Storage place	26.9%	26.5%		72.6%	72.3%		0.2%	1.2%		0.3%					
(storage period; day)	(2.2)	(3.6)		(13.0)	(18.5)		(2.0)	(2.7)		(8.5)					
Shellfish			0.012			0.003			0.987		-	0.158		-	0.014
Storage place	37.4%	30.5%		59.3%	67.5%		2.0%	2.0%		0.3%			1.0%		
(storage period; day)	(2.5)	(3.2)		(11.4)	(15.4)		(2.2)	(2.9)		(8.5)			(-)		
Fruit and vegetables			0.000			0.000			0.002		-	0.158	-	-	-
Storage place	96.9%	90.7%		0.3%	3.1%		2.5%	6.1%		0.3%					
(storage period; day)	(5.7)	(7.8)		(7.0)	(4.1)		(3.6)	(5.1)		(5.0)					
Eggs			0.000	-		0.000			0.009	-	-	-	-	-	-
Storage place	96.9%	90.6%			3.1%		3.1%	6.3%							
(storage period; day)	(11.7)	(14.5)			(8.8)		(13.3)	(12.8)							
Milk			0.000	-		0.000			0.472	-	-	-	-	-	-
Storage place	99.5%	95.2%			4.0%		0.5%	0.8%							
(storage period; day)	(3.5)	(5.8)			(5.0)		(2.3)	(6.4)							
Frozen processed foods			0.003			0.027	-		0.003	-	-	-		-	0.000
Storage place	5.1%	9.6%		92.6%	88.9%			1.5%					2.3%		
(storage period; day)	(7.8)	(16.0)		(14.7)	(23.2)			(16.2)					(-)		

^1^*p*-value was provided according to the results of the Kruskal–Wallis test conducted for the analysis on the differences of responses in each answer option (only for storage place, not for storage period) between Year 2010 and Year 2019.

**Table 5 foods-09-01457-t005:** Consumer behaviors for food purchasing, preparing, cooking, eating, leftovers management, and kitchen utensils.

Questions and Choices	Percent of Respondents or Answers		*p*-Value ^1^
	Year 2010 (N = 609)	Year 2019 (N = 605)	
Q13. Do you thaw only a portion of food as much as you intend to cook rather than thawing all of the foods?			
Never	1.3%	1.0%	0.6
Seldom	3.9%	3.8%	0.9
Sometimes	10.8%	19.0%	0
Frequently	50.2%	45.1%	0.074
Always	33.2%	31.1%	0.435
Do not know/no response	0.5%	–	0.084
<Analysis of the combined responses> ^2^			
Never + Seldom	5.3%	4.8%	0.713
Always + Frequently	83.4%	76.2%	0.002
Q14. Do you use plastic gloves when you handle meat, fish/shellfish, or eggs?			
Never	8.9%	6.1%	0.069
Seldom	16.4%	21.2%	0.035
Sometimes	17.6%	22.5%	0.033
Frequently	30.0%	30.6%	0.841
Always	27.1%	19.7%	0.002
<Analysis of the combined responses> ^2^			
Never + Seldom	25.3%	27.3%	0.432
Always + Frequently	57.1%	50.2%	0.016
Q15. What do you do when you have a slight wound on your hand during cooking?			
Keep cooking after the wound has been treated	74.7%	71.7%	0.242
Keep cooking without any treatment	19.7%	24.6%	0.039
Do not cook	5.6%	3.6%	0.018
Q17. Do you wash fruits and vegetables before eating?			
Never	0.3%	0.5%	0.649
Seldom	0.3%	1.8%	0.012
Sometimes	1.8%	12.6%	0
Frequently	30.2%	30.4%	0.94
Always	67.2%	54.7%	0
<Analysis of the combined responses> ^2^			
Never + Seldom	0.7%	2.3%	0.017
Always + Frequently	97.4%	85.1%	0
Q20. Do you store remaining raw foodstuffs with sealing after cooking?			
Never	0.3%	0.3%	0.995
Seldom	2.3%	2.6%	1
Sometimes	3.9%	16.4%	0
Frequently	46.3%	39.3%	0.014
Always	47.0%	41.3%	0.048
<Analysis of the combined responses> ^2^			
Never + Seldom	2.6%	3.0%	0.713
Always + Frequently	93.3%	80.7%	0

^1^*p*-value was provided according to the results of the Kruskal–Wallis test conducted for the analysis on the differences of responses in each answer option between Year 2010 and Year 2019. ^2^ To identify the difference between positive responses (i.e., Always + Frequently) and negative responses (i.e., Never + Seldom), answer options were combined.

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
