# Peer review of "A Closer Look at Changes in High-Risk Food-Handling Behaviors and Perceptions of Primary Food Handlers at Home in South Korea across Time"

_foods, 2020, doi:10.3390/foods9101457_

Round 1
Reviewer 1 Report
A Closer Look at Changes in Risky Food-handling Behaviors and Perceptions of Primary Food Handlers at Home in South Korea across Time
Thank you for taking the time to address my comments from the first round of review and for the opportunity to re-review this manuscript. The quality of this submission has improved significantly from the initial review, however, there are still issues that I believe should be considered prior to publication. My main concern is the focus on perception versus actual behaviour. It is well recognised that surveys cannot be used as a measure of actual behaviour, rather self-reported behaviours and this is not acknowledged in the literature. Prior to publication I would recommend that the authors address the following points.
General comments
- Questionnaires alone cannot reveal data on actual domestic food handling behaviours as they are only measures of self-reported behaviours. To explore actual behaviours methods such as ethnographic observation in the home would be necessary and there is a body of literature on this. Line 87-88 is therefore difficult for a quantitative study to substantiate.
- The novelty of the study is identified to be the fact that the same study was implemented over two time periods, what would be interesting to explore is was there any educational intervention or social marketing campaigns during this period in South Korea that may have influenced levels of knowledge over time? To me the interesting thing would be to explore the potential influences on behaviour between the two time points in addition to identifying persistently risky reported practice. In addition, it would be interesting for the research to explore differences in knowledge/behaviour profiles of respondents and conducted a segmentation analysis to support targeted FS education. This might be for a second paper but it would be useful to highlight.
- It would be useful for the introduction for the reader to be presented with more contextual information regarding the burden of foodborne disease in South Korea, you mention that a very small proportion originated in the domestic environment, however the number of overall cases is not reported. Reporting the overall number of cases/deaths and economic burden of this would better justify and contextualise need for the research.
- In the absence of domestic food safety guidelines in South Korea the questionnaire was developed based on the guidelines of the US, why were the guidelines of the US chosen as the framework for the study? Why were these chosen over the guidance’s in the UK for example?
- Section 2.2 it would be useful for this section to provide some details on any of the changes made to the survey following piloting.
- Section 3.2 it is not clear to me how Figure 1 shows large gaps between consumers’ food safety perceptions and actual behaviours in surveys performed in 2010 and 2019. As I highlight in comment 1 questionnaire data can only report on self-reported rather than actual behaviours and it is therefore difficult to use this instrument to draw conclusions around gaps in perception and actual behaviour. What I think this does show is potential for deviant behaviour. I suggest that you look at the following papers that explore this.
- McCarthy, M., Brennan, M., Kelly, A.L., Ritson, C., De Boer, M. and Thompson, N., 2007. Who is at risk and what do they know? Segmenting a population on their food safety knowledge. Food Quality and Preference, 18(2), pp.205-217.
- De Boer, M., McCarthy, M., Brennan, M., Kelly, A.L. and Ritson, C., 2005. Public understanding of food risk issues and food risk messages on the island of Ireland: the views of food safety experts. Journal of food safety, 25(4), pp.241-265.
- Kendall, H., Brennan, M., Seal, C., Ladha, C. and Kuznesof, S., 2016. Behind the kitchen door: a novel mixed method approach for exploring the food provisioning practices of the older consumer. Food Quality and Preference, 53, pp.105-116.
I see that you have asked a perception based question in line with guidance then a behavioural question as a follow up, to explore perception against self-reported behaviours. This needs to be clearly articulated in section 2.1 and the narrative supporting Table 1, although be clear that a survey instrument can only provide information on self-reported kitchen practice and not actual kitchen practice as I note above.
- Line 262: Another example that should state that consumers “reported to practice proper food handling in line with guidance”
- Line 181: states “consumer still did not wash perishables in Year 2019” why would behaviour change between the years? Again, as I comment above, what factors might cause difference in response between the two years? I am not clear what the value of comparison is with consideration of factors that may influence response?
- Line 183-184: It is not immediately obvious that the 51.7% in Year 2010 to 35.7% in Year 2019 for ‘completely yes 184 (always) + mostly yes (frequently)’ is the combined value of the responses to each question in the respective years, either make this clearer in the text or annotate the figure.
- Line 208-209 also refers to the gap between perceptions and actual behaviour, I think you mean potential for deviant behaviour, see comments above.
- Moderate response to questions highlights potential for behaviour that deviates from best practice guidelines- this warrants discussion and comparison with the findings of the extant body of literature in the field, again see examples above.
- Line 227-229: the subheading in the table under head “topic” are not consistent with those listed in the text above, amend to make these consistent with the titles used in the table as these are more suitable.
- Line 334-336: Needs to be re-worded, it is not clear what is meant by this and is not a good opening to the discussion.
- Throughout the results you refer to the “proper risk perception” do you mean the identification of the correct response in line with domestic food safety guidance? Consider revising this or at least explaining more clearly what is meant my “proper risk perception” which is slightly confusing to the reader.
- Throughout the reporting of the results the scale completely yes, moderate, completely no is used when explaining the findings, this is not a very well recognised scale and it would be more appropriate to use the frequency scale that is included in brackets in Table 1. This would be more familiar to readers.
Edits:
Line 47: should the number 4 be here? “1.32% 4 outbreaks and 47 0.36% cases were attributed to foods prepared at private home”
Line 47: Ministry of Food and Drug Sfoafety should this be safety?
Line 57-60: Should this be “can be used to explore” rather than figure out?
Domestic food safety might be a better term to use throughout the paper.
Line 63: change researches to research
Line 79: Change to “Behind risky food behaviour”
Line 83: change to previous research.
Line 95-98: The following sentence need editing and should be amended to read:
To develop questionnaires, experts in different fields, representing the South Korean government (Ministry of Food and 96 Drug Safety, Cheongju-si, Korea), food safety laboratories (Korea University, Seoul, Korea), consumer organizations, and a professional market research company (Gallup Korea, Seoul, Korea) formed a consulting committee.
Line 98: change: composed to compiled
Line 105: should this be levels of behaviour or just hazardous behaviour?
Table 1 Q2 and Q5 should this be “at room temperature” rather than “under room temperature”? you might not be able to change this if this is how the question was asked to participants although it is grammatically incorrect.
Table 1 Q8: remove “the” from the question to make this grammatically correct, although as above if this is how the questions were asked it might be difficult to change.
Table 1 Q7-9 why are there two response options? Always-never works better for these questions.
Table 1 Q11- Same as above?
Q12: change “thawing” to thaw
Q15 change to “keep cooking after the wound has been treated”
Q14-16, Q16-17 and Q19 the always –never scale would be more appropriate here
Q 18 Change to: Store at room temperature
Q21 Change others to other
Line 115-117 reorder the sentence to start with “To confirm the clarity of draft questionnaire a pilot test, using a draft questionnaire, was conducted with 15 randomly…”
Line 119: change “that is” to that are
Line 143: Does this mean the average duration of the data collection process was approximately 3 months? If so, I think this could be written more clearly.
Line 188: at instead of under?
Line 189: at room temperature instead of in
Line 198-199: increased instead of “even were increased”
Line 200: “worsened between studies” instead of got worse during 8 years
Line 221-223: clear sate that this was a difference between survey rounds.
Line 235: Change sentence to Common risk food handling practices of consumers …
Line 272-274: This is a very confusing sentence revise. “Responses to proper risk perceptions” do you mean that XX percent of participants identified the correct response in line with food safety guidance? Whilst XX percent did not and this was a difference of XX% between 2010 and 2019?
Line 235: Delete “were”, line 235 is an unfinished sentence?
Line 278: Change to: Most participants perceived there…
Line 279:Change to self-reported behaviours
Line 281: confusing sentence revise, do you mean that although there were decreases observed the majority of participants identified the correct food handling practice and reported to practice this at home?
Line 287-292 is a very long and confusing sentence, I would suggest that you simplify, by addressing point 15 would help here.
Line 297: Change to recommend
Line 297: you refer to Survey 1 and 2 here but in the rest of the manuscript by date Year 2010 and 2019, be consistent throughout, survey 1 and 2 is more concise but if you change to this you need to make sure that you reference this in the methods section.
Line 334-367: change “researches” to research
Line : Change to “Previous consumer surveys regarding the home food safety identify the risk perception and behaviors in the specific time-point.
Line 352: This study does not provide a cross cultural comparison as suggested here but adds to the body of literature regarding domestic food handling practices in different nations.
Line 372 doesn’t make sense can this be re-worded?
Line 380-383: Behaviours are unlikely to change over time unless there is an intervention,
Line 384-387: this sentence doesn’t make sense, can it be re-worded? What is “clean” this should be defined.
Line 388-389, deviation in knowledge versus practice observed, see references suggested in point 5 above.
Line 403: at room temperature not under
Line 407-408: Are you sure?
Line 435: remove “the” from “the both risk”
Line 438: A recent report
Line 442: Change to: Also reported consumer preference for washing utensils in hot water with detergent
Line 450: “highlights”
Line 464-465: Segmentation analysis is necessary to identify consumers that demonstrate high levels of knowledge regarding best practice but tendencies to report deviant behaviour and those with low knowledge and therefore likelihood to also demonstrate deviant behaviours see: Kendall, H., Kuznesof, S., Seal, C., Dobson, S. and Brennan, M., 2013. Domestic food safety and the older consumer: A segmentation analysis. Food quality and preference, 28(1), pp.396-406.
Author Response
Response to Reviewer 1 Comments
Author response: Thank you for providing us helpful comments. We checked the manuscript with the perspective of your comments and revisions were highlighted as “yellow” in the submitted manuscript.
Point 1: Thank you for taking the time to address my comments from the first round of review and for the opportunity to re-review this manuscript. The quality of this submission has improved significantly from the initial review, however, there are still issues that I believe should be considered prior to publication. My main concern is the focus on perception versus actual behaviour. It is well recognised that surveys cannot be used as a measure of actual behaviour, rather self-reported behaviours and this is not acknowledged in the literature. Prior to publication I would recommend that the authors address the following points. Questionnaires alone cannot reveal data on actual domestic food handling behaviours as they are only measures of self-reported behaviours. To explore actual behaviours methods such as ethnographic observation in the home would be necessary and there is a body of literature on this. Line 87-88 is therefore difficult for a quantitative study to substantiate.
Response 1: Thank you for the helpful comment regarding the self-reported behaviors. To follow your comment, we replaced the statements and/or expressions describing the “actual behaviors” to “self-reported behaviors” or “the potential for the deviate behaviors”. We highlighted that the responses to the questionnaires regarding the consumers’ behaviors implied the potential for the deviate behaviors because those responses are self-reported. We added the statements regarding this issue in the INTRODUCTION section to prevent the misunderstanding of consumers with the perspectives to the ‘actual behaviors’.
Point 2: The novelty of the study is identified to be the fact that the same study was implemented over two time periods, what would be interesting to explore is was there any educational intervention or social marketing campaigns during this period in South Korea that may have influenced levels of knowledge over time? To me the interesting thing would be to explore the potential influences on behaviour between the two time points in addition to identifying persistently risky reported practice. In addition, it would be interesting for the research to explore differences in knowledge/behaviour profiles of respondents and conducted a segmentation analysis to support targeted FS education. This might be for a second paper but it would be useful to highlight.
Response 2: As you recommended, we agree that the research to explore differences in knowledge/behaviour profiles of respondents and to conduct the segmentation analysis for targeted FS education will be suitable as a second paper. Since we cannot demonstrate the factor because we did not consider the procedures for the identification of the determinant factor during the survey, we added the statements regarding the importance of the further examination of the determinant factors which can induce the changes in consumers’ risk perception and/or behaviors in the CONCLUSION section in the revised manuscript to highlight this issue.
Point 3: It would be useful for the introduction for the reader to be presented with more contextual information regarding the burden of foodborne disease in South Korea, you mention that a very small proportion originated in the domestic environment, however the number of overall cases is not reported. Reporting the overall number of cases/deaths and economic burden of this would better justify and contextualise need for the research.
Response 3: Thank you for the comment. In the case of the number of overall outbreak and cases of domestic foodborne illnesses, we did not indicate actual values because those data were assumed to be significantly underestimated (i.e. the number of both outbreaks and cases are less than 10). Because of this limitation, we think that the indication of the actual number will not be helpful for the justification of the need for the research. Rather, we highlighted that the actual number of outbreak and cases in South Korea might also be much higher because previous relevant researchers assumed that most foodborne illnesses were unreported and/or unconfirmed.
Point 4: In the absence of domestic food safety guidelines in South Korea the questionnaire was developed based on the guidelines of the US, why were the guidelines of the US chosen as the framework for the study? Why were these chosen over the guidance’s in the UK for example?
Response 4: We chose the guidelines of the US FDA because the domestic food safety guidelines for consumers have been provided in various forms (e.g. documents, online web pages, post in social network services, pamphlets) and have been consistently revised. We did not come up with the plan for the consideration of guidelines from various countries. Please excuse this issue.
Point 5: Section 2.2 it would be useful for this section to provide some details on any of the changes made to the survey following piloting.
Response 5: Thank you for the comment. Since the questionnaires in the present study were in Korean, most revisions after the piloting are also the problems of the expressions in Korean. So, we concluded that the results of the piloting will be better to be emitted from this manuscript to prevent the confusion of potential international readers of this manuscript. Please excuse this issue.
Point 6: Section 3.2 it is not clear to me how Figure 1 shows large gaps between consumers’ food safety perceptions and actual behaviours in surveys performed in 2010 and 2019. As I highlight in comment 1 questionnaire data can only report on self-reported rather than actual behaviours and it is therefore difficult to use this instrument to draw conclusions around gaps in perception and actual behaviour. What I think this does show is potential for deviant behaviour. I suggest that you look at the following papers that explore this.
McCarthy, M., Brennan, M., Kelly, A.L., Ritson, C., De Boer, M. and Thompson, N., 2007. Who is at risk and what do they know? Segmenting a population on their food safety knowledge. Food Quality and Preference, 18(2), pp.205-217.
De Boer, M., McCarthy, M., Brennan, M., Kelly, A.L. and Ritson, C., 2005. Public understanding of food risk issues and food risk messages on the island of Ireland: the views of food safety experts. Journal of food safety, 25(4), pp.241-265.
Kendall, H., Brennan, M., Seal, C., Ladha, C. and Kuznesof, S., 2016. Behind the kitchen door: a novel mixed method approach for exploring the food provisioning practices of the older consumer. Food Quality and Preference, 53, pp.105-116.
I see that you have asked a perception based question in line with guidance then a behavioural question as a follow up, to explore perception against self-reported behaviours. This needs to be clearly articulated in section 2.1 and the narrative supporting Table 1, although be clear that a survey instrument can only provide information on self-reported kitchen practice and not actual kitchen practice as I note above.
Response 6: To follow your comment, we revised the expressions regarding the “actual behaviors” in the overall manuscript to follow your comment. We added the statements regarding this issue in the INTRODUCTION section to prevent the misunderstanding of consumers with the perspectives to the ‘actual behaviors’. We also newly added all the suggested references above to clarify that the responses of the behaviors were self-reported and thus the results of the analysis of the survey data implied the potential for deviant behaviors.
Point 7: Line 262: Another example that should state that consumers “reported to practice proper food handling in line with guidance”
Response 7: We added the explanation you suggested. Thank you.
Point 8: Line 181: states “consumer still did not wash perishables in Year 2019” why would behaviour change between the years? Again, as I comment above, what factors might cause difference in response between the two years? I am not clear what the value of comparison is with consideration of factors that may influence response?
Response 8: To follow your comment, we modified the sentences in the revised manuscript to avoid the expressions of direct comparison between Year 2010 and 2019. In the case of the factor which is expected to cause difference in response between the two years, now we cannot demonstrate the factor because we did not consider the procedures for the identification of the determinant factor during the survey (also stated in the Response 2 of this round of the revision).
Point 9: Line 183-184: It is not immediately obvious that the 51.7% in Year 2010 to 35.7% in Year 2019 for ‘completely yes 184 (always) + mostly yes (frequently)’ is the combined value of the responses to each question in the respective years, either make this clearer in the text or annotate the figure.
Response 9: Thank you for the comment. To follow your comment, we added the annotation about the indication of the location of the data regarding the combined values (Supplementary tables) in the titles of figures.
Point 10: Line 208-209 also refers to the gap between perceptions and actual behaviour, I think you mean potential for deviant behaviour, see comments above.
Response 10: We revised the expressions regarding the “actual behaviors” in the overall manuscript to follow your comment (also as shown in the Response 6 above).
Point 11: Moderate response to questions highlights potential for behaviour that deviates from best practice guidelines- this warrants discussion and comparison with the findings of the extant body of literature in the field, again see examples above.
Response 11: As you commented, moderate response can be regarded as the potential for behavior that deviates from best practice guidelines. To discuss and compare the results of this study with the body of literature in this field, we think we should obtain the additional information especially for the intention of the survey participants who responded the moderate responses and thus we are planning to newly conduct the further survey research by contacting the respondents for the study of the present manuscript. For the revision of this manuscript with the perspectives to the ‘moderate response’, we added the statements regarding the impact of the in-depth examination about the moderate respondents in the DISCUSSION section with the citation of the recommended references above to highlight the necessity of the further research on the investigation of the moderate response.
Point 12: Line 227-229: the subheading in the table under head “topic” are not consistent with those listed in the text above, amend to make these consistent with the titles used in the table as these are more suitable.
Response 12: According to your comment, we amended to make “topics” in the text above Table 3 consistent with the titles used in Table 3.
Point 13: Line 334-336: Needs to be re-worded, it is not clear what is meant by this and is not a good opening to the discussion.
Response 13: Thank you for helpful comment. We simplified and clarified the meaning of the sentence in the revised manuscript.
Point 14: Throughout the results you refer to the “proper risk perception” do you mean the identification of the correct response in line with domestic food safety guidance? Consider revising this or at least explaining more clearly what is meant my “proper risk perception” which is slightly confusing to the reader.
Response 14: To prevent the confusion of readers, we added the explanation regarding the meaning of the “proper risk perception” (i.e. the correct response in line with conventional food safety guidelines) and “improper risk perception” (i.e. the response in opposition to conventional food safety guidelines) at the first use of those expressions.
Point 15: Throughout the reporting of the results the scale completely yes, moderate, completely no is used when explaining the findings, this is not a very well recognised scale and it would be more appropriate to use the frequency scale that is included in brackets in Table 1. This would be more familiar to readers.
Response 15: Thank you for the suggestion. We replaced the scale “completely yes-completely no” to “Never-Always” in the revised manuscript.
Point 16: Edits:
Line 47: should the number 4 be here? “1.32% 4 outbreaks and 47 0.36% cases were attributed to foods prepared at private home”
Line 47: Ministry of Food and Drug Sfoafety should this be safety?
Line 57-60: Should this be “can be used to explore” rather than figure out?
Domestic food safety might be a better term to use throughout the paper.
Line 63: change researches to research
Line 79: Change to “Behind risky food behaviour”
Line 83: change to previous research.
Line 95-98: The following sentence need editing and should be amended to read:
To develop questionnaires, experts in different fields, representing the South Korean government (Ministry of Food and Drug Safety, Cheongju-si, Korea), food safety laboratories (Korea University, Seoul, Korea), consumer organizations, and a professional market research company (Gallup Korea, Seoul, Korea) formed a consulting committee.
Line 98: change: composed to compiled
Line 105: should this be levels of behaviour or just hazardous behaviour?
Table 1 Q2 and Q5 should this be “at room temperature” rather than “under room temperature”? you might not be able to change this if this is how the question was asked to participants although it is grammatically incorrect.
Table 1 Q8: remove “the” from the question to make this grammatically correct, although as above if this is how the questions were asked it might be difficult to change.
Table 1 Q7-9 why are there two response options? Always-never works better for these questions.
Table 1 Q11- Same as above?
Q12: change “thawing” to thaw
Q15 change to “keep cooking after the wound has been treated”
Q14-16, Q16-17 and Q19 the always –never scale would be more appropriate here
Q 18 Change to: Store at room temperature
Q21 Change others to other
Line 115-117 reorder the sentence to start with “To confirm the clarity of draft questionnaire a pilot test, using a draft questionnaire, was conducted with 15 randomly…”
Line 119: change “that is” to that are
Line 143: Does this mean the average duration of the data collection process was approximately 3 months? If so, I think this could be written more clearly.
Line 188: at instead of under?
Line 189: at room temperature instead of in
Line 198-199: increased instead of “even were increased”
Line 200: “worsened between studies” instead of got worse during 8 years
Line 221-223: clear sate that this was a difference between survey rounds.
Line 235: Change sentence to Common risk food handling practices of consumers …
Line 272-274: This is a very confusing sentence revise. “Responses to proper risk perceptions” do you mean that XX percent of participants identified the correct response in line with food safety guidance? Whilst XX percent did not and this was a difference of XX% between 2010 and 2019?
Line 235: Delete “were”, line 235 is an unfinished sentence?
Line 278: Change to: Most participants perceived there…
Line 279:Change to self-reported behaviours
Line 281: confusing sentence revise, do you mean that although there were decreases observed the majority of participants identified the correct food handling practice and reported to practice this at home?
Line 287-292 is a very long and confusing sentence, I would suggest that you simplify, by addressing point 15 would help here.
Line 297: Change to recommend
Line 297: you refer to Survey 1 and 2 here but in the rest of the manuscript by date Year 2010 and 2019, be consistent throughout, survey 1 and 2 is more concise but if you change to this you need to make sure that you reference this in the methods section.
Line 334-367: change “researches” to research
Line : Change to “Previous consumer surveys regarding the home food safety identify the risk perception and behaviors in the specific time-point.
Line 352: This study does not provide a cross cultural comparison as suggested here but adds to the body of literature regarding domestic food handling practices in different nations.
Line 372 doesn’t make sense can this be re-worded?
Line 380-383: Behaviours are unlikely to change over time unless there is an intervention,
Line 384-387: this sentence doesn’t make sense, can it be re-worded? What is “clean” this should be defined.
Line 388-389, deviation in knowledge versus practice observed, see references suggested in point 5 above.
Line 403: at room temperature not under
Line 407-408: Are you sure?
Line 435: remove “the” from “the both risk”
Line 438: A recent report
Line 442: Change to: Also reported consumer preference for washing utensils in hot water with detergent
Line 450: “highlights”
Line 464-465: Segmentation analysis is necessary to identify consumers that demonstrate high levels of knowledge regarding best practice but tendencies to report deviant behaviour and those with low knowledge and therefore likelihood to also demonstrate deviant behaviours see: Kendall, H., Kuznesof, S., Seal, C., Dobson, S. and Brennan, M., 2013. Domestic food safety and the older consumer: A segmentation analysis. Food quality and preference, 28(1), pp.396-406.
|
Response 16: All edits suggested from you were applied in the revised manuscript. Responses for specific comments which needed an additional explanation regarding the revision are as follows:
- Line 272-274: This is a very confusing sentence revise. “Responses to proper risk perceptions” do you mean that XX percent of participants identified the correct response in line with food safety guidance? Whilst XX percent did not and this was a difference of XX% between 2010 and 2019?
à Response: To follow your comment, we revised the sentence.
- Line 235: Delete “were”, line 235 is an unfinished sentence?
à Response: We deleted “were suggested” and added “could be identified” to make line 235 as finish sentence.
- Line 281: confusing sentence revise, do you mean that although there were decreases observed the majority of participants identified the correct food handling practice and reported to practice this at home?
à Response: To prevent the confusion of readers, we deleted that sentence.
- Line 297: you refer to Survey 1 and 2 here but in the rest of the manuscript by date Year 2010 and 2019, be consistent throughout, survey 1 and 2 is more concise but if you change to this you need to make sure that you reference this in the methods section.
à Response: Actually we prefer the expression ‘survey 1’ and ‘survey 2’, but revised those expressions to ‘Year 2010’ and ‘Year 2019’ according to the comments from the reviewer in the process of the 2nd So we decided to follow the reviewer’s comments, and thus used the expressions ‘Year 2010’ and ‘Year 2019’. We didn’t realize that there were expressions of ‘survey 1 and 2’ in the manuscript, and thus we replaced the ‘surveys 1 and 2’ to ‘Year 2010 and Year 2019’.
- Line 352: This study does not provide a cross cultural comparison as suggested here but adds to the body of literature regarding domestic food handling practices in different nations.
à Response: The “survey researches” in the original version of the manuscript means the “previous research” and we revised the terms in that sentence to highlight this issue.
- Line 372 doesn’t make sense can this be re-worded?
à Response: To follow your comment, we revised the sentence.
- Line 380-383: Behaviours are unlikely to change over time unless there is an intervention
à Response: To follow your comment, we deleted the expression “One interesting finding was that” from the original version of the manuscript.
- Line 384-387: this sentence doesn’t make sense, can it be re-worded? What is “clean” this should be defined.
à Response: “Clean” is one of the topics (i.e. Clean, Separate, Cook, Chill) for the 4 basic steps for food safety at home suggested from FDA. We agree that this term may confuse the readers, so we replaced the term “Clean” to cleaning in the revised manuscript.
- Line 407-408: Are you sure?
à Response: We concluded that there have been no research highlighted the risk perception-behavior gaps of food consumers by focusing on the changes in those gaps over time. To follow your comment, however, we used the relative expression (i.e. rarely reported) in the revised manuscript to avoid using the definite expression (i.e. have not been reported) in the original version of the manuscript.
- Line 464-465: Segmentation analysis is necessary to identify consumers that demonstrate high levels of knowledge regarding best practice but tendencies to report deviant behaviour and those with low knowledge and therefore likelihood to also demonstrate deviant behaviours see: Kendall, H., Kuznesof, S., Seal, C., Dobson, S. and Brennan, M., 2013. Domestic food safety and the older consumer: A segmentation analysis. Food quality and preference, 28(1), pp.396-406.
à Response: We agree your opinion for the necessity of the segmentation analysis. Thus, we added the statements (with the citation of the reference you recommended) explaining the importance of segmentation analysis for the identification of major education targets to narrow the risk perception-behavior gaps.
We want to say that we acknowledge helpful discussion and valuable comments.
Reviewer 2 Report
In the document the authors use the work "risky." This should be changed to High Risk, or Low Risk or Moderate Risk, etc. For example, in the title the authors write "Risky Food Handling Behaviors" This wording should be changed to "High Risk Food Handling Behaviors." Similar changes need to be made throughout the manuscript.
Author Response
Response to Reviewer 2 Comments
Author response: Thank you for providing us helpful comments. We checked the manuscript with the perspective of your comments and revisions were highlighted as “light-blue” in the submitted manuscript.
Point 1: In the document the authors use the work "risky." This should be changed to High Risk, or Low Risk or Moderate Risk, etc. For example, in the title the authors write "Risky Food Handling Behaviors" This wording should be changed to "High Risk Food Handling Behaviors." Similar changes need to be made throughout the manuscript.
Response 1: In this study, the word "risky" means the "high risk". To follow your comment, we changed the word "risky" used in the overall manuscript to "high risk". Thank you for the helpful comment.
We want to say that we acknowledge helpful discussion and valuable comments.
This manuscript is a resubmission of an earlier submission. The following is a list of the peer review reports and author responses from that submission.
Round 1
Reviewer 1 Report
GENERAL COMMENTI
The topic is interesting and relevant.
However, there are some several critical aspects.
- The fundamental critical aspect is that the statistical treatment is completely lacking. How can authors conclude that there is/there is not a difference in responses between 2010 and 2019? Only based on %?
- Authors did not explore at all how important variables like age, location, or level of instruction, etc. differently impacted on the knowledge of safty issue and safety perception, therefore data seem be treated in a very superficial way;
- Authors must clearly specify the answers options for each question in the questionnaire (the scales) in Table 1 and the statistical analysis must be added and specified.
- All verbal anchors in Table 6 seem used in an incorrect way: frequencies should have been asked (like never, sometimes...) instead of agreement Completely no, Mostly no, Moderate, Mostly yes, Completely yes (which reminds to a Likert scale).
MAJOR
- Authors must specify in abstracts that the survey was conducted on different subjects from 2010 and 2019
- Line 62 Define among brackets what is meant for “primary food handlers”. “consumers at home”?
- line 62 authors should remove the word “longitudinal” because the subjects were not the same.
- Introduction lacks of a paragraph related to the main risk perceptions at home, since this is also a topic of the questionnaire, a paragraph on this should be added.
- Table 1. It is necessary to specify which were the options of each answer and how many points each scale had. To improve readability authors can add a column to Tab. 1 with answer options.
- Which options of answer had the Question: Q. Where do you store each food (meat, chicken, fish, shellfish, fruit/vegetables, eggs, milk, and frozen processed foods)?
- line 97: it is not clear how participants were selected and contacted. Briefly explain “pre-allocated”.
- Table 1 and Table 2 should be merged into a unique table and a chi squared test could be applied to show that the samples were comparable (no effect of the year was found in the population composition).
- data analysis is completely lacking. Authors should briefly specify which analyses were conducted
- Table 6: for Q. When you thaw meat, fish, and shellfish, only portion for cooking was thawed? The options of answers are not congruent with the question or English is not clear.
- In Table 6. The anchors of the many scales is used in an incorrect way (Completely no, Mostly no, Moderate, Mostly yes, Completely yes à frequencies could have been asked like never, sometimes...).
- In Table. 6 at Q. How do you sanitize cutting board?, Q. How do you sanitize kitchen cloth? an important option is lacking which is the “no sanitization options are applied”.
MINOR
Uniform the verbs at the correct forms (past) in abstract and in the text
Author Response
Response to Reviewer 1 Comments
Author response: Thank you for providing us helpful comments. We checked the manuscript with the perspective of your comments and revisions were highlighted as “green” in the submitted manuscript. Revisions by the common comments from all reviewers (reviewer #1, #2, and #3) and an academic editor were highlighted as “yellow”.
Point 1: [General comment 1] The fundamental critical aspect is that the statistical treatment is completely lacking. How can authors conclude that there is/there is not a difference in responses between 2010 and 2019? Only based on %?
Response 1: According to your comment, we conducted the statistical analysis which can be represented by pearson chi-square test and Kruskal-Wallis test for the evaluation on the differences in respondents’ composition for each sociodemographic characteristic (i.e. a variable) and the responses for each answer option to the survey questions between the time-points of the survey (2010 and 2019), respectively. Statements regarding the differences in responses between 2010 and 2019 were based on the statistical analysis in the overall contents of the revised manuscript.
Point 2: [General comment 2] Authors did not explore at all how important variables like age, location, or level of instruction, etc. differently impacted on the knowledge of safty issue and safety perception, therefore data seem be treated in a very superficial way.
Response 2: Thank you for the comment. We strongly agree with your opinion that the complementation of descriptive statistics by the statistical analysis for the investigation regarding the impact of variables to the survey results can be supportive to the findings and implication of this research. However, unfortunately, this study was originally designed to identify the distinct gap between the consumers’ risk perception and practices obviously observed regardless of the time-point (survey 1 in 2010 and survey 2 in 2019). Thus the distribution of the populations in each variable of sociodemographic characteristics was not optimally determined to analyse the extent of the importance of the variables (e.g. age, location, education level, etc.). Rather our findings suggested the background information for further studies to design in-depth analysis on the identification of major determinant factors on the phenomenon observed in this research (e.g. gap between risk perception and behaviors). Thus, to follow your comment, now we are planning to conduct the further research by performing surveys optimally designed for the impact of various variables. We think that our results are important to be the representative case as the reports of underrecognized phenomenon linked to the food safety at home. Please excuse this issue.
Point 3: [General comment 3] Authors must clearly specify the answers options for each question in the questionnaire (the scales) in Table 1 and the statistical analysis must be added and specified.
Response 3: We tried to include answer options in the Table 1, but it made that Table too complicated. Thus, we did not include answer options for each question in the Table 1 and clearly showed all answer options in the Figures and Tables from the RESULTS section. Rather we added more footnotes in major Tables including Table 1 (Contents of questionnaire) to provide supportive information for the answer options.
Point 4: [General comment 4] All verbal anchors in Table 6 seem used in an incorrect way: frequencies should have been asked (like never, sometimes...) instead of agreement Completely no, Mostly no, Moderate, Mostly yes, Completely yes (which reminds to a Likert scale).
Response 4: We agree that the expression of the answer options in Table 6 should have been in the form of frequencies. Respondents were received the explanation regarding the meaning of those answer options (Completely no, Mostly no, Moderate, Mostly yes, Completely yes) with the perspective to the frequencies. To follow your comment, we added the actual meaning for each answer option (Completely no, Mostly no, Moderate, Mostly yes, Completely yes) not only in Table 6 but also other similar cases (i.e. Figure 1 and 2).
Point 5: [Major 1] Authors must specify in abstracts that the survey was conducted on different subjects from 2010 and 2019
Response 5: Following your comment, we specified that the survey was conducted on different subjects from 2010 and 2019 in ‘ABSTRACT’.
Point 6: [Major 2] Line 62 Define among brackets what is meant for “primary food handlers”. “consumers at home”?.
Response 6: The meaning of “primary food handlers” is ‘food consumers who are the main people involved in food preparation at home’, and this explanation used in the manuscript is now provided in the brackets for the term “primary food handlers” (firstly shown) in the INTRODUCTION section.
Point 7: [Major 3] line 62 authors should remove the word “longitudinal” because the subjects were not the same.
Response 7: We removed the word “longitudinal” to follow your comment.
Point 8: [Major 4] Introduction lacks of a paragraph related to the main risk perceptions at home, since this is also a topic of the questionnaire, a paragraph on this should be added.
Response 8: According to your advices, we added a new paragraph regarding the risk perception which can affect the consumers’ behaviors in the INTRODUCTION section. Based on that background information, we also emphasized the importance of understanding both consumers’ risk perception and relevant risky behaviors.
Point 9: [Major 5] Table 1. It is necessary to specify which were the options of each answer and how many points each scale had. To improve readability authors can add a column to Tab. 1 with answer options.
Response 9: Thank you for the comment. We tried to add a column to Table 1 with answer options according to your comment, but we concluded that the repetition of same answer options for multiple questions might cause low readability. Thus, we provided statements regarding the answer options by adding footnotes in Table 1 as described in the response to the Point 3 from your comment.
Point 10: [Major 6] - Which options of answer had the Question: Q. Where do you store each food (meat, chicken, fish, shellfish, fruit/vegetables, eggs, milk, and frozen processed foods)?
Response 10: We added the detailed explanation regarding the options of answer for the question ‘Q. Where do you store foods (meat, chicken, fish, shellfish, fruit/vegetables, eggs, milk, and frozen processed foods)?’ and ‘Q. How long do you store foods (meat, chicken, fish, shellfish, fruit/vegetables, eggs, milk, and frozen processed foods) in each storage place?’ in the footnote of Table 1 to improve the readability following your comment.
Point 11: [Major 7] - line 97: it is not clear how participants were selected and contacted. Briefly explain “pre-allocated”.
Response 11: “Pre-allocated” means the method of the selection of respondents (i.e. quota sampling). Sampling fraction and sample size according to each sociodemographic characteristic described in this study were determined by referring the official statistical population data in South Korea. We contacted the respondents by the face-to-face interview conducted in hoseholds or shopping center. Participants of the survey were asked for their sociodemographic characteristics to obtain responses according to the pre-allocated population composition. Explanation regarding the method of the selection and contact of the participants is now added in section “2.3. Survey” (MATERIALS AND METHODS section) in the revised manuscript.
Point 12: [Major 8] - Table 1 and Table 2 should be merged into a unique table and a chi squared test could be applied to show that the samples were comparable (no effect of the year was found in the population composition).
Response 12: According to your comment, Table 1 and Table 2 were merged into a single Table 2 in the revised manuscript. Then the results of the statistical analysis (chi squared test) using the population composition were also added to this merged Table (Table 2 in the revised manuscript; Sociodemographic characteristics of the respondents and chi-square test results for each variable). As we expected, results of the statistical analysis evaluated no effect of the year for all variables (sociodemographic characteristics) considered in this study [i.e. gender, age (years), location, level of education, number of family members]. We stated the aim and the results of the statistical analysis for the population composition in the revised manuscript.
Point 13: [Major 9] data analysis is completely lacking. Authors should briefly specify which analyses were conducted
Response 13: Statistical analyses conducted in this study are pearson chi-square test and Kruskal-Wallis test. We added the statements regarding the data analysis method in the revised manuscript for both MATERIALS AND METHODS (section “2.4. Data analysis”) and RESULTS.
Point 14: [Major 10] Table 6: for Q. When you thaw meat, fish, and shellfish, only portion for cooking was thawed? The options of answers are not congruent with the question or English is not clear.
Response 14: To clarify the meaning of the question, we revised the statement for the question from “When you thaw meat, fish, and shellfish, only portion for cooking was thawed?” to “Do you thaw only portion of foods (e.g. meat, fish, and shellfish) as much as you intend to cook rather than thawing the whole of foods?”.
Point 15: [Major 11] In Table 6. The anchors of the many scales is used in an incorrect way (Completely no, Mostly no, Moderate, Mostly yes, Completely yes), frequencies could have been asked (like never, sometimes...).
Response 15: Answer options were revised to follow your comment. For more detailed information regarding this issue, please see the response 4 to your comment point 4.
Point 16: [Major 12] In Table. 6 at Q. How do you sanitize cutting board?, Q. How do you sanitize kitchen cloth? an important option is lacking which is the “no sanitization options are applied”.
Response 16: Thank you for the comment. Since the lack of the option “no sanitization options are applied” may confuse readers, we deleted these questions (Q. How do you sanitize cutting board?, Q. How do you sanitize kitchen cloth?) and following answers in the revised manuscript. According to your comment, we believe that this deletion can improve the readability for readers.
Point 17: [Minor 1] Uniform the verbs at the correct forms (past) in abstract and in the text.
Response 17: We checked the overall manuscript including the ABSTRACT and the text to change the form of verbs (past) to follow your comment. Thank you.
We want to say that we acknowledge helpful discussion and valuable comments.

Reviewer 2 Report
Foods
A Closer Look at Changes in Risky Food-handling Behaviors and Perceptions of Primary Food Handlers at Home in South Korea across Time
I make the following observations to justify my position:
- The paper fails to provide any justification for why exploring the food handling practices of primary food handlers in South Korean households is necessary i.e. what is the significance of study? The introduction begins by presenting an overview of microbiological food borne disease incidence in the US and Europe (context) although provides no indication of the situation in South Korea to contextualise and justify this research.
- No overall research questions or aims of the research are identified.
- No consideration of the existing body of literature in this area. There is a very large literature which has not been considered, failing to identify any of the known factors that influence consumer’s food handling knowledge or practice.
- Significant methodological issues including:
- No clear justification for sampling self-reported food handling practices over two time periods is offered.
- The questionnaire developed based on domestic food handling guidelines of the USA, not clear why these were not the recommendations specific to South Korea. This could introduce significant bias to the research as households may well have no knowledge of US recommendations/may not be culturally applicable. Unless the survey was based on these owing to an absence of specific South Korean domestic food handling recommendations? (which could be justification for the study) Or these were adapted for the SK context? this is not made clear.
- Heavily female biased sample.
- Different sample populations used in the first and second survey making any time comparisons flawed.
- No indication of the data analysis approach taken, SPSS used but what method is unclear.
- Findings show that there were gaps between perceptions and actual behaviours and no changes between surveying rounds, however, it is not clear to the reader what has occurred during these time periods that is of significance (if anything). For example, have there been communication campaigns? Policies? Etc. what factors would influence change and how have these been tested other than running a survey twice with different populations?
- Figures used to present the data are small and difficult to interpret analysis, analysis is crude, using only descriptive statistics and percentage responses, no statistical significant presented or more detailed exploration of factors influencing food safety behaviours which is suggested in the introduction (i.e. age, gender, education level, vulnerability etc.)
- Discussion is descriptive providing limited critical analysis of the research findings and no consideration given to how the findings this research compare with studies in this domain, of which there are many.
- Requires improvements to writing style and language to be made throughout, i.e. the first line does not make grammatical sense.
Author Response
Response to Reviewer 2 Comments
Author response: Thank you for providing us helpful comments. We checked the manuscript with the perspective of your comments and revisions were highlighted as “light-blue” in the submitted manuscript. Revisions by the common comments from all reviewers (reviewer #1, #2, and #3) and an academic editor were highlighted as “yellow”.
Point 1: The paper fails to provide any justification for why exploring the food handling practices of primary food handlers in South Korean households is necessary i.e. what is the significance of study? The introduction begins by presenting an overview of microbiological food borne disease incidence in the US and Europe (context) although provides no indication of the situation in South Korea to contextualise and justify this research.
Response 1: Thank you for your comment. Consumer food handling behaviours is very important to prevent foodborne disease. Investigating consumers’ behaviors and perceptions at home is essential for establishment of the proper and practical guidelines and countermeasures. Thus, researchers in various countries mainly in the Western countries reported behaviors and perceptions of consumers at home with consumer survey (Kennedy et al., 2005; Murray et al., 2017; Patil, Cates, & Morales, 2005; Röhr, Lüddecke, Drusch, Müller, & Alvensleben, 2005; Terpstra, Steenbekkers, De Maertelaere, & Nijhuis, 2005; Verbeke, Sioen, Pieniak, Van Camp, & De Henauw, 2005). However, there is only a limited number of studies in South Korea. Significance of this study is that the present study provided comprehensive data from a nationwide survey which could be used as a basic information to enhance household food safety awareness. Another interesting approach of the study was the investigation by trend study on a decade basis (2010 and 2019) with same questionnaires for both surveys. Most previous researches as we mentioned earlier were mainly based on the cross-sectional approaches without considering the changes in time frameworks. Since consumers’ behaviors and perceptions are to be altered by food trends over time, the analysis of key changes can support the design of effective interventions against improperly changed behaviors with risk perceptions. To make clear our opinion, we revised the INTRODUCTION section with providing proper references as below:
<Reference>
Kennedy, J., Jackson, V., Blair, I., McDowell, D., Cowan, C., & Bolton, D. (2005). Food safety knowledge of consumers and the microbiological and temperature status of their refrigerators. Journal of Food Protection, 68, 1421-1430.
Murray, R., Glass-Kaastra, S., Gardhouse, C., Marshall, B., Ciampa, N., Franklin, K., Hurst, M., Thomas, M. K., & Nesbitt, A. (2017). Canadian consumer food safety practices and knowledge: Foodbook study. Journal of Food Protection, 80, 1711-1718.
Patil, S. R., Cates, S., & Morales, R. (2005). Consumer food safety knowledge, practices, and demographic differences: findings from a meta-analysis. Journal of Food Protection, 68, 1884-1894.
Röhr, A., Lüddecke, K., Drusch, S., Müller, M. J., & Alvensleben, R. (2005). Food quality and safety––consumer perception and public health concern. Food Control, 16, 649-655.
Terpstra, M., Steenbekkers, L., De Maertelaere, N., & Nijhuis, S. (2005). Food storage and disposal: consumer practices and knowledge. British Food Journal.
Verbeke, W., Sioen, I., Pieniak, Z., Van Camp, J., & De Henauw, S. (2005). Consumer perception versus scientific evidence about health benefits and safety risks from fish consumption. Public Health Nutrition, 8, 422-429.
In addition, for clear justification of the present study, indications of the situation in South Korea were added while several sentences describing an overview of microbiological foodborne disease incidence in the US and Europe were deleted in the revised manuscript.
<Added sentences>
“In South Korea, it is assumed that the number of foodborne outbreak and cases were underreported; Ministry of Food and Drug Safety reported that only 1.32% 4 outbreaks and 0.36% cases were attributed to foods prepared at private home (Ministry of Food and Drug Safety, 2020).”
“However, there has been only a limited number of studies in South Korea.”
<Removed sentences>
“The Centers for Disease Control and Prevention (CDC) reported that 48 million cases of foodborne illness occur; more than 128,000 people are hospitalized, and 3,000 people die each year in the United States.”
“In the case of Europe, approximately one-third of foodborne diseases occurred at home.”
Point 2: No overall research questions or aims of the research are identified.
Response 2: The statements for the major topics of the research questions and the aims of this consumer survey study are now added in the revised manuscript. Since the previous researches on consumers’ risk perception and/or risky behaviors regarding the home food safety have been conducted by the cross-sectional analyses, we expected that unchanged and/or emerging risky behaviors could be identified by the comparative analysis of two individual nation-wide surveys with same questionnaires over a decade. We think that our results reported through the research paper can be the representative case for the analysis of multiple individual surveys feasible to figure out the demonstration of the distinct/obvious gap between risk perception and practices regardless of the time-point for the surveys. We emphasized this issue at the final paragraph of INTRODUCTION section in the revised manuscript with the statement for defined motivation and study design: possible novelty and/or contribution of the manuscript to the research areas regarding the consumers’ structure of the manuscript according to the major aims.
Point 3: No consideration of the existing body of literature in this area. There is a very large literature which has not been considered, failing to identify any of the known factors that influence consumer’s food handling knowledge or practice.
Response 3: Thank you for the comment. For the literature review which should have been placed in both ‘INTRODUCTION’ and ‘DISCUSSION’ section, we newly analysed the recently reported relevant references and added results and/or implications regarding the major risks in consumers’ food handling knowledge (risk perception) with practical behaviors in the revised manuscript. We suggested the list of recent major researches regarding home food safety in INTRODUCTION section, and the analyses for those literatures were provided in the DISCUSSION section. Especially we added the results of comparative analysis with other literatures at the end of each paragraph in DISCUSSSION section to suggest the major implications and findings.
Point 4: Significant methodological issues including: a. No clear justification for sampling self-reported food handling practices over two time periods is offered.
Response 4: To follow your comment, we added the explanation regarding the method for obtaining consumers’ responses from both surveys (survey 1 in 2010 and survey 2 in 2019) at MATERIALS AND METHODS section “2.3. Surveys” in the revised manuscript.
Point 5: Significant methodological issues including: b. The questionnaire developed based on domestic food handling guidelines of the USA, not clear why these were not the recommendations specific to South Korea. This could introduce significant bias to the research as households may well have no knowledge of US recommendations/may not be culturally applicable. Unless the survey was based on these owing to an absence of specific South Korean domestic food handling recommendations? (which could be justification for the study) Or these were adapted for the SK context? this is not made clear.
Response 5: As we mentioned ‘2.1. Questionnaires for food-handling behaviors and perception’ in MATERIALS AND METHODS section from the original version of the manuscript, questionnaires were developed by a consulting committee with experts in different fields including Korea government (Ministry of Food and Drug Safety: major institution responsible for food safety in Korea), consumer organization, and professional survey company. At that time, there was no established guideline provided by the Korea government. The aim of this grant was to set the food handling guideline to improve the in-house food hygiene and safety management. We tried to search general food safety guidelines for the home provided by health authorities such as US FDA and USDA and adopted them to develop draft of questionnaires. Only general guidelines were included to a draft questionnaire while guidelines not culturally applicable to Korean consumer were excluded. Draft of questionnaires inspected by a consulting committee were revised according to their opinion. We revised the ‘2.1. Questionnaires for food-handling behaviors and perception’ in MATERIALS AND METHODS section to avoid misunderstanding.
Point 6: Significant methodological issues including: c. Heavily female biased sample.
Response 6: Thank you for the comment. We unavoidably used heavily female biased sample because of the distinct distribution of male and female primary food handlers at home in South Korea [background information (reports from Korean press: https://www.hankookilbo.com/News/Read/201801172130873058]. We also think that the consideration of the recent changes in the increases of male food handlers in South Korea is likely to derive the interesting findings which can act as a clue of the underreported and/or emerging risky behaviors from male food handlers. Thus, this limitation of this research and proposals for future research were stated in the CONCLUSION section in the revised manuscript.
Point 7: Significant methodological issues including: d. Different sample populations used in the first and second survey making any time comparisons flawed.
Response 7: We agree with your concern regarding the comparative analysis by using responses of different subjects from two individual surveys. Thus, we mainly focused on the distinct and obvious trends observed in both surveys for the gap in risk perception and actual behaviors of consumers. Comparison between responses from survey 1 and 2 was also supplemented by statistical analysis following major comments from reviewers in this revision, however, we agree your comment that direct comparison for same respondents and two time periods can effectively show the impact of time in the risky behaviors and perception of consumers. Whereas since the aim of this research is to obtain general responses from consumers (i.e. representative sample) in each survey, we recruited the respondents not only from survey 1 (2010) but also survey 2 (2019) to use a representative sample in each time-point of the survey. This issue regarding the limitation of this research and proposals for future research were stated in the CONCLUSION section in the revised manuscript.
Point 8: Significant methodological issues including: e. No indication of the data analysis approach taken, SPSS used but what method is unclear.
Response 8: Statistical analyses conducted in this study are pearson chi-square test and nonparametric Kruskal-Wallis test. We added the statements regarding the data analysis method in the revised manuscript for both MATERIALS AND METHODS (section 2.4. Data analysis) and RESULTS section.
Point 9: Findings show that there were gaps between perceptions and actual behaviours and no changes between surveying rounds, however, it is not clear to the reader what has occurred during these time periods that is of significance (if anything). For example, have there been communication campaigns? Policies? Etc. what factors would influence change and how have these been tested other than running a survey twice with different populations?
Response 9: Thank you for the suggestion. We mentioned the opinions for this issue in the DISCUSSION section in the original version of the manuscript: “The South Korean government has tried to improve food safety (e.g. advertising, providing guidelines and publishing pamphlets); however, this study found the difficulty in the correction of consumers’ behaviors.” [this sentence is now revised as follows: “The South Korean governmental institutions governing home food safety have tried to improve risk perceptions and practices of consumers (e.g. advertising, providing guidelines, publishing pamphlets, providing education, etc.); however, this study found the difficulty in the correction of consumers’ behaviors.”]. There has been various kinds of the intervention strategies adopted by the South Korean government (e.g. advertising, providing guidelines, publishing pamphlets, providing education, etc.) during the survey periods of this study (2010~2019), but we did not focus on the specific strategy. The aim of this research was to obtain general responses from general consumers (i.e. representative sample of primary food handlers in South Korean) in each survey, and thus we could not analyse the impact of specific intervention strategies. To obtain the valuable results suggested from your comment, we will start consumer survey project as the further study of this research to evaluate the effectiveness of various intervention strategies to the gap of risk perception and behaviors observed by our findings. Since we agree with your concern on the confusion of readers, we added the statement regarding this issue in CONCLUSION section. Please excuse this issue.
Point 10: Figures used to present the data are small and difficult to interpret analysis, analysis is crude, using only descriptive statistics and percentage responses, no statistical significant presented or more detailed exploration of factors influencing food safety behaviours which is suggested in the introduction (i.e. age, gender, education level, vulnerability etc.)
Response 10: To improve the clarity of the Figures, we changed all figures with higher resolution than original version and increases the size of those newly added figures in the revised manuscript. Moreover, to improve the readability, Figure 2 in the original version of the manuscript was replaced by the Table 3 in the revised manuscript. Figure files were also attached in the submission system of MDPI and thus the high-resolution images can be published if this manuscript is accepted. In the case of the detailed exploration of factors, we carefully tried to suggest the findings from the impact of variables to the results of this study, but we concerned about the limitation of this study with the perspective to the factor analysis. We strongly agree with your opinion that the complementation of descriptive statistics by the statistical analysis for the investigation regarding the impact of variables to the survey results can be supportive to the findings and implication of this research. However, unfortunately, this study was originally designed to identify the distinct gap between the consumers’ risk perception and practices obviously observed regardless of the time-point (survey 1 in 2010 and survey 2 in 2019). Thus the distribution of the populations in each variable of sociodemographic characteristics is not optimal to analyse the extent of the importance of the variables (e.g. age, location, education level, etc.). Rather our findings suggest the background information for further studies to design in-depth analysis of the identification of major determinant factors on the phenomenon observed in this research (e.g. gap between risk perception and behaviors). Thus, to follow your comment, now we are planning to conduct the further research by performing surveys optimally designed for the impact of various variables. We considered the impact of the major variables only by the analysis of the association between each variable of sociodemographic characteristics and the year (please see Table 2 in the revised manuscript). Please excuse this issue.
Point 11: Discussion is descriptive providing limited critical analysis of the research findings and no consideration given to how the findings this research compare with studies in this domain, of which there are many.
Response 11: To highlight the reason why our findings are important to consider in the fields of consumer behaviour studies with the perspectives to food safety issues, DISCUSSSION section is now entirely revised by adding the results of comparative analysis with other literatures at the end of each paragraph which describes the major implications.
Point 12: Requires improvements to writing style and language to be made throughout, i.e. the first line does not make grammatical sense.
Response 12: Since other reviewers and the academic editor also required the improvement of writing style and language for the clarity, we accepted those requirements and revised overall contents of the manuscript after the application of the responses to the reviewers’ comments. We also carefully revised the first line as you suggested.
We want to say that we acknowledge helpful discussion and valuable comments.

Reviewer 3 Report
Good information, and a comprehensive review of consumer practices. However, in the discussion section, inclusion by the authors on practical suggestions for proper handling, and storage of food at the home is needed and would be helpful. For example, although it is not a recommended practice, to thaw foods at room temperature, food can be thawed at home under cold water as long as the consumer is monitoring the amount of time needed to thaw the product. Foodborne hazards are time/temperature dependent. In addition, in the Discussion Section, the authors should discuss the gaps in consumer practices such as the importance of proper maintenance, cleaning and sanitizing of cutting boards. It would be helpful and informative, if the authors would include a few sentences describing practical ways that consumers can address these gaps.
The authors state that most consumers do not wash or trim food prior to storage. The authors should provide examples of the types of products that require washing and trimming and those that do not, and why and why not certain type food require washing and trimming and others may not. Additional practical information would be helpful describing how consumers can properly refrigerate and/or freeze large food items besides cutting them into smaller pieces. Include suggestions on proper packaging, how not to group large amount of food together, suggestions on how to separate large pieces placing them into individual packages as an alternative to cutting them into smaller pieces, how to place them properly in the freezer to ensure optimum quick freezing, etc.
Author Response
Response to Reviewer 3 Comments
Author response: Thank you for providing us helpful comments. We checked the manuscript with the perspective of your comments and revisions were highlighted as “grey” in the submitted manuscript. Revisions by the common comments from all reviewers (reviewer #1, #2, and #3) and an academic editor were highlighted as “yellow”.
Point 1: Good information, and a comprehensive review of consumer practices. However, in the discussion section, inclusion by the authors on practical suggestions for proper handling, and storage of food at the home is needed and would be helpful. For example, although it is not a recommended practice, to thaw foods at room temperature, food can be thawed at home under cold water as long as the consumer is monitoring the amount of time needed to thaw the product. Foodborne hazards are time/temperature dependent. In addition, in the Discussion Section, the authors should discuss the gaps in consumer practices such as the importance of proper maintenance, cleaning and sanitizing of cutting boards. It would be helpful and informative, if the authors would include a few sentences describing practical ways that consumers can address these gaps.
Response 1: Thank you for the positive evaluation of this manuscript and helpful comments. To follow your comments, we suggested the additional information at DISCUSSION section in the revised manuscript as follows: 1) practical suggestions for proper handling and storage of foods at the home, 2) gaps in consumer practices such as the importance of proper maintenance, cleaning and sanitizing of cutting boards. We also added various new references to provide the knowledge sources to ensure food safety at home by primary food handlers.
1) practical suggestions for proper handling and storage of foods at the home: We added the explanations and representative references regarding the proper methods for food handlers in all paragraphs describing the implications from the results of this study [i.e. total four paragraphs for large gaps of perception-behaviors observed in this study as follows: 1) storing perishable foods without any preparation (washing or trimming), 2) thawing foods at room temperature, 3) exposing leftovers to danger zone temperatures, and 4) failing to separate kitchen utensils such as cutting boards and kitchen cloths by use.]. In the case of practical suggestion on the thawing foods under cold water at home, we highlighted the importance of time-temperature control as the prerequisite to follow your comment.
2) gaps in consumer practices such as the importance of proper maintenance, cleaning and sanitizing of cutting boards: To narrow the gap between the management of kitchen utensils including the cutting boards, we added the major implications from the previous relevant researches with appropriate references cited in the revised manuscript as follows: cross-contamination pathways from raw materials to cooked foods, the preference on various methods for the management of cutting boards, and effective intervention methods against pathogens by decontamination of cutting boards.
Point 2: The authors state that most consumers do not wash or trim food prior to storage. The authors should provide examples of the types of products that require washing and trimming and those that do not, and why and why not certain type food require washing and trimming and others may not. Additional practical information would be helpful describing how consumers can properly refrigerate and/or freeze large food items besides cutting them into smaller pieces. Include suggestions on proper packaging, how not to group large amount of food together, suggestions on how to separate large pieces placing them into individual packages as an alternative to cutting them into smaller pieces, how to place them properly in the freezer to ensure optimum quick freezing, etc.
Response 2: Thank you for the comment. We added the practical information regarding the home food safety in DISCUSSION section according to your comment as follows: 1) the examples of the types of products that require washing and trimming and those that do not, and why and why not certain type food require washing and trimming, 2) how consumers can properly refrigerate and/or freeze large food items besides cutting them into smaller pieces.
1) the examples of the types of products that require washing and trimming and those that do not, and why and why not certain type food require washing and trimming: Since the most important food products to be washed and/or trimmed before storage or consumption are fruit and vegetables eaten as raw, we provided the additional explanation and effective methods for
the preparation of those food items (i.e. washing and trimming) in DISCUSSION section as follows: “The importance for washing and trimming of foods especially for eaten as raw (e.g. fruits, vegetables, etc.) has been highlighted by the previous researches on the effectiveness of the decontamination of microbiological risk factors (i.e. pathogens) and/or the validation of the cross-contamination.”.
2) how consumers can properly refrigerate and/or freeze large food items besides cutting them into smaller pieces: Although we tried to find out the effective guidelines or research reports regarding this issue, we concluded that further researches have to be conducted to suggest consumers (i.e. primary food handlers) the standard intervention methods. So, we mentioned this limitation on the knowledge basis in DISCUSSION section and provided the desirable examples suggested from you (i.e. Suggestions on proper packaging, how not to group large amount of food together, suggestions on how to separate large pieces placing them into individual packages as an alternative to cutting them into smaller pieces, how to place them properly in the freezer to ensure optimum quick freezing).
We want to say that we acknowledge helpful discussion and valuable comments.

Round 2
Reviewer 1 Report
Results are still too exposed in an unclear way. Authors did not add systematically a comment on significant differences between 2010 and 2019: results from Kruskal-Wallis test method were only mentioned in data analysis but not shown nor commented in results. Commenting in detail each percentage is not useful if it is not specified whether the trend significantly changed or not from 2010 and 2019 (paragraphs: 3.1.1.2, 3.1.1.3., 3.1.1.4. etc..).
Chi-square test could provide overall differences in distributions of answers between 2010 and 2019 (while Kruskal-Wallis test differences for each answer’s modality). In general, I wish that authors first comment overall differences between the two years (2010 vs 2019) and then specific differences for each answer’s modality.
Below specific comments.
MAJOR
- Results in abstract could be re-written to better clarify results. In fact, it is a unclear: 1. which incorrect behaviours characterized the two years (2010, 2019); 2. which are the were significant differences between the two years (in other words did behaviours improved or got worst?). An hypothesis to rephrase could be: “Year 2010 was characterized by.... Year 2019 by ... The trends during 9 years improved/get worst for the following aspects...”.
- Lines 17- 19: Do the results mentioned in lines 17-19 referred to year 2010 or to 2019 or to both years? Please specify.
- Table 1: to improve readability, the verbal anchors to the questionnaire should be answers in a column right to the table. In my opinion, sending back to the figures does not improve readability.
- line 106: it is not a ‘subjective question’ but rather “an open question”.
- Talking about “survey 1” and “survey 2” is misleading since the survey was the same. Authors should rather talk about “2010” and “2019”. Replace this in lines 158-161 and in the text.
- In table 1 and in figures: rename the questions with increasing numbers (Q1, Q2, etc.).
- Results: it is still not easy to capture from figures if the distribution from 2010 was significantly differen from 2019. Moreover, where are summarized the results of Kruskal-Wallis test method mentioned in lines 159-160 for each question? Authors should add significant asterisks on all figures. Consider adding a new table (like Tab. 2) for all questions.
- Results: in general not all results of all questions are shown in figures. This is confusing.
- A major concern remain since authors inserted in data analysis Kruskal-Wallis test method but did not comment at all the results based on significance difference between the two years!!! basically, authors just added the analysis in the data analysis but did not comment critically the data based on significant difference. This can be seen in paragraph 3.1.1.2, 3.1.1.3., 3.1.1.4. etc..
-Table 3 helps in better understanding the results but why did authors insert only a few questions? p.value should be uniformed in the number of decimal digits.
- Why two different colors are used (red, green) to comments % in all figures? Maybe for positive or negative behaviours? Add in the legends of all figures.
- Are results from tables 4 and 5 significantly different? This comment is lacking. To compare the results from 2010 and 2019 authors could have applied a Chi-square test on % of results. This approach could have been used also in the other questions.
MINOR
- In paragraph 2.3. Surveys specify the number of total questions which composed the questionnaire and the avarage duration.
- Line 15: change “consumer surveys for different subjects” into “consumer surveys from different subjects”
- Line 99: insert commas as follows: “draft questionnaire while guidelines, not culturally applicable to Korean consumers, were excluded
- Line 106: “wee” instead of “are”
Author Response
Response to Reviewer 1 Comments
Author response: Thank you for providing us helpful comments. We checked the manuscript with the perspective of your comments and revisions were highlighted as “green” in the submitted manuscript.
Point 1: Results are still too exposed in an unclear way. Authors did not add systematically a comment on significant differences between 2010 and 2019: results from Kruskal-Wallis test method were only mentioned in data analysis but not shown nor commented in results. Commenting in detail each percentage is not useful if it is not specified whether the trend significantly changed or not from 2010 and 2019 (paragraphs: 3.1.1.2, 3.1.1.3., 3.1.1.4. etc..).
Chi-square test could provide overall differences in distributions of answers between 2010 and 2019 (while Kruskal-Wallis test differences for each answer’s modality). In general, I wish that authors first comment overall differences between the two years (2010 vs 2019) and then specific differences for each answer’s modality.
Response 1: Thank you for the helpful comment. According to your comment, RESULTS section was revised especially for the contents linked to Figure 1 and 2 because of the lack of the explanation and expressions regarding the statistical analysis in the 1st revision. We agree with your opinion, “authors should (1) first comment overall differences between the two years (2010 vs 2019) and (2) then specific differences for each answer’s modality”. Firstly, in the case of (1) comment on overall differences between the two years, we tried to use the statistical analysis method as chi-square test. But we worried that the results of the overall differences in distribution of answers (which could be interpreted by chi-square test) could be considerably affected by the changes in the responses to an answer option, “moderate” (i.e. the significant differences in the distribution of the responses could be shown by changes in the responses to “moderate” even if there was no noticeable changes in the negative and positive responses). In the case of Figure 1 (risk perception-behavior gap) and Figure 2 (Proper perception and behaviors), we wanted to emphasize the “overall differences between two years (2010 and 2019)” with the perspective to “distinct differences in the relationship of risk perception and behaviors”. Figure 1 highlighted that consumers knew safe methods for food-handling (proper risk perception), but actual behaviors did not support their confidence in food safety (improper behavior) (To clarify this definition of the ‘risk perception-behavior gap’ according to your comment, results on the kitchen utensils were deleted from this manuscript because the distinct differences in the proper and improper behavior were not observed). To effectively explain this issue to readers, we had to induce readers to focus on the distinct trends in the extent of the percentages for the responses to all answer options for both questions to risk perception and the behavior. Figure 2 highlighted that consumers knew safe methods for food-handling (proper risk perception), and actual behaviors supported their confidence in food safety (proper behavior). As mentioned above (i.e. the issue on Figure 1), to effectively explain this issue to readers, we also had to induce readers to focus on the distinct trends in the extent of the percentages for the responses to all answer options for both questions to risk perception and the behavior. So we stated the (1) overall differences between two years by the explanation on the distinct and unchanged trends for the extent of the percentages for the responses to all answer options rather than the explanation on the significant differences on the distribution of responses (from the statistical analysis of chi-square test). Please excuse this issue. Secondly, in the case of (2) comment on specific differences for each answer’s modality, the results of the statistical analysis (Kruskal-Wallis test) were added in the RESULTS section. We appreciate your comment which can improve the manuscript.
Point 2: [Major comment 1] Results in abstract could be re-written to better clarify results. In fact, it is a unclear: 1. which incorrect behaviours characterized the two years (2010, 2019); 2. which are the were significant differences between the two years (in other words did behaviours improved or got worst?). An hypothesis to rephrase could be: “Year 2010 was characterized by.... Year 2019 by ... The trends during 9 years improved/get worst for the following aspects...”.
Response 2: ABSTRACT is now revised to emphasize the differences between the two years by using the expressions (improved, got worst, improvement, etc.) as you suggested. We also tried to apply the hypothesis to rephrase described in Point 2. Moreover, to increase the readability, we changed the terms for surveys from survey 1 and survey 2 to Year 2010 and Year 2019, respectively (according to the Response 6).
Point 3: [Major comment 2] Lines 17- 19: Do the results mentioned in lines 17-19 referred to year 2010 or to 2019 or to both years? Please specify.
Response 3: We specified the relationship between the results and the time-point of the survey (in 2010).
Point 4: [Major comment 3] Table 1: to improve readability, the verbal anchors to the questionnaire should be answers in a column right to the table. In my opinion, sending back to the figures does not improve readability.
Response 4: To follow your comment, we added all answer questions by adding a new column right in the Table 1.
Point 5: [Major comment 4] line 106: it is not a ‘subjective question’ but rather “an open question”.
Response 5: We replaced the expression ‘subjective question’ into ‘an open question’ to follow your comment.
Point 6: [Major comment 5] Talking about “survey 1” and “survey 2” is misleading since the survey was the same. Authors should rather talk about “2010” and “2019”. Replace this in lines 158-161 and in the text.
Response 6: According to your comment, we revised the overall contents in the revised manuscript regarding talking about “survey 1” and “survey 2”. Rather we used the expressions for describing the survey conducted in 2010 and the survey conducted in 2019 as “Year 2010” and “Year 2019”, respectively (these expressions were adopted from your comment 2). We also defined those terms “Year 2010” and “Year 2019” in the INTRODUCTION section.
Point 7: [Major comment 6] In table 1 and in figures: rename the questions with increasing numbers (Q1, Q2, etc.).
Response 7: We renamed the questions with increasing numbers in Table 1 and those numbers were also indicated in results (Figure 1, Table 3, Figure 2, Table 4, Table 5, Table 6).
Point 8: [Major comment 7] Results: it is still not easy to capture from figures if the distribution from 2010 was significantly differen from 2019. Moreover, where are summarized the results of Kruskal-Wallis test method mentioned in lines 159-160 for each question? Authors should add significant asterisks on all figures. Consider adding a new table (like Tab. 2) for all questions.
Response 8: To follow your comment, we added the significant asterisks on all figures (Figure 1, Figure 2). In the case of the addition of a new table for all questions, we added Supplementary Tables to avoid the repetition of the results data in the manuscript. Since we considered that using Figures is desirably adequate to describe the results of risk perception-behavior gap (Section 3.1.) and proper behaviors (Section 3.3.), Figures were used in the manuscript and the detailed data were provided as Supplementary Tables.
Point 9: [Major comment 8] Results: in general not all results of all questions are shown in figures. This is confusing.
Response 9: We checked the overall data and revised the Figures. Moreover, since we added the number of questions (according to the Response 7) and all answer options (according to the Response 4) in Table 1, we expected to demonstrate that all results were inserted in the manuscript.
Point 10: [Major comment 9] A major concern remain since authors inserted in data analysis Kruskal-Wallis test method but did not comment at all the results based on significance difference between the two years!!! basically, authors just added the analysis in the data analysis but did not comment critically the data based on significant difference. This can be seen in paragraph 3.1.1.2, 3.1.1.3., 3.1.1.4. etc..
Response 10: Thank you for the comment. In the case of 1st response, we did not added the comments regarding the significance differences between the two years in Section 3.1. and Section 3.3.. We considered that the comments regarding the significant differences to discuss the risk perception-behavior gap and proper behaviors were not needed because those gap and proper behaviors were distinct trends which could be observed regardless of the time-point of surveys (2010 for survey 1, 2019 for survey 2). However, as you suggested, the results regarding the statistical analysis should have been indicated in both figures and statements (RESULTS, DISCUSSION section). To follow your comment, we revised the figures, added the supplementary Tables to provide detailed results of statistical analysis, and added the comments in both RESULTS (Section 3.1. and 3.3.) & DISCUSSION sections.
Point 11: [Major comment 10] Table 3 helps in better understanding the results but why did authors insert only a few questions? p.value should be uniformed in the number of decimal digits.
Response 11: We revised the manuscript by following your comments:
(1) Insertion of few questions: According to the responses from consumers, we categorized the results into ‘Discordance between consumers’ food safety perceptions and actual behaviors (Risk perception-behavior gap)’ (Section 3.1.), ‘Changes in common risky behaviors between surveys’ (Section 3.2.), and ‘Proper behaviors of consumers’ (Section 3.3.). All questions in the questionnaire were inserted in the manuscript as Figure 1, Table 3, Figure 2, Table 4, Table 5, or Table 6. Since the number of each question was added in the revised manuscript (according to the Response 7), we expected to demonstrate that all results were inserted in the manuscript.
(2) decimal digits of p-value: We uniformed the number of decimal digits in the revised manuscript.
Point 12: [Major comment 11] Why two different colors are used (red, green) to comments % in all figures? Maybe for positive or negative behaviours? Add in the legends of all figures.
Response 12: Two different colors represented by green and red were used to indicate the positive (proper) and negative (risky) perceptions/behaviors as you commented. To improve the readability, we added the legends of all figures.
Point 13: [Major comment 12] Are results from tables 4 and 5 significantly different? This comment is lacking. To compare the results from 2010 and 2019 authors could have applied a Chi-square test on % of results. This approach could have been used also in the other questions.
Response 13: Thank you for the comment. To follow your comment, we revised the structure of the Tables (i.e. Tables 4 and 5 were combined into Table 4) for the addition of the results for the statistical analysis in the Table. In the case of the method for the statistical analysis, Kruskal-Wallis test was used since there have been distinct trends in the preference on the storage place for each food item. We had to emphasize the distinct preferences (i.e. meat for freezer, chicken for refrigerator or freezer, fish for freezer, shellfish for freezer, fruit and vegetables for refrigerator, eggs for refrigerator, milk for refrigerator, and frozen processed foods for freezer) to let readers focus on those distinct trends although there was the significant differences in the distribution of the responses between Year 2010 and Year 2019. We worried that the results of chi-square test could confuse readers if we highlighted both unchanged trends in the distribution of the preference (interpreted by the level of the percentages for answer options) and significant differences in the distribution of the preference (interpreted by the chi-square test). Please excuse this issue.
Point 14: [Minor comment 1] In paragraph 2.3. Surveys specify the number of total questions which composed the questionnaire and the avarage duration.
Response 14: To follow your comment, the number of total questions in the questionnaire and the average duration for the surveys were now indicated in the paragraph 2.3. in the revised manuscript by adding sentences regarding these issues.
Point 15: [Minor comment 2] Line 15: change “consumer surveys for different subjects” into “consumer surveys from different subjects”
Response 15: The preposition ‘for’ was changed into ‘from’ to follow your comment: “consumer surveys for different subjects” à “consumer surveys from different subjects”.
Point 16: [Minor comment 3] Line 99: insert commas as follows: “draft questionnaire while guidelines, not culturally applicable to Korean consumers, were excluded
Response 16: As you suggested, we inserted commas: “draft questionnaire while guidelines, not culturally applicable to Korean consumers, were excluded”.
Point 17: [Minor comment 4] Line 106: “wee” instead of “are”
Response 17: We followed your comment: “are” à “were”.
We want to say that we acknowledge helpful discussion and valuable comments.
